# Evolution of iGluR ligand specificity, polyamine regulation, and ion selectivity inferred from a placozoan epsilon receptor
Anhadvir Singh[1], Boris S. Zhorov [2,3,9], Luis A. Yanez-Guerra [4,5,9], Alessandra Aleotti [6,9], Chloe C. Koens [7,9], C. Defne Yanartas[1], Yunqi Song[1], Federico Javier Miguez Cabello[8], Derek Bowie [8] & Adriano Senatore [1] ✉

Epsilon ionotropic glutamate receptors (iGluRs) are a recently defined clade of neurotransmitter receptors that are found in all major metazoan lineages that are distinct from α-amino-3-hydroxy-5-methyl-4-isoxazolepropionic acid (AMPA), kainate, delta, phi (i.e., AKDF) and *N*-methyl-D-aspartate NMDA receptors. Here, we explore the evolution of iGluRs by generating a broad species-guided phylogeny of eukaryotic iGluRs and a comprehensive phylogeny of placozoan receptors, uncovering marked diversification of epsilon type receptors within Placozoa. Functional characterization of one epsilon receptor from the placozoan species *Trichoplax adhaerens*, named GluE1αA, reveals sensitivity to glycine, alanine, serine, and valine, but not glutamate. We demonstrate that changing just three amino acids in the ligand binding domain could convert ligand specificity of GluE1αA from glycine to glutamate, also causing nascent sensitivity to AMPA and increased sensitivity to the blocker 6-cyano-7-nitroquinoxaline-2,3-dione (CNQX). We also demonstrate that an atypical serine in the pore Q/R/N site confers diminished $Ca^{2+}$ permeation and sensitivity to polyamine block, imposing similar effects on the human GluA2 receptor, and that a conserved aspartate four amino acids downstream of the Q/R/N site is crucial for polyamine regulation. Thus, key molecular determinants for polyamine regulation are conserved between AKDF and epsilon receptors.

Ionotropic glutamate receptors (iGluRs) belong to a large and diverse family of ligand-gated ion channels present in eukaryotes and prokaryotes[1,2]. The most studied of these are the vertebrate NMDA, AMPA and kainate receptors, collectively known for their important roles in excitatory synaptic signaling in the mammalian brain[3,4]. Recently, a landmark phylogenetic study defined two novel clades of metazoan iGluRs, the epsilon receptors that originated very early during animal evolution and are found broadly in bilaterians and non-bilaterians, and lambda receptors that are exclusive to sponges[5]. This study also revealed that AMPA and kainate receptors belong to a larger group that also includes Delta and Phi receptors (hence collectively named AKDF receptors), and that NMDA receptors are absent in non-bilaterian animals except cnidarians. Several more recent studies have reported that sponge lambda receptors are phylogenetically proximal to

plant receptors rather than other metazoan iGluRs[1,6,7]. To date, the functional properties of lambda receptors have not been reported, but epsilon receptors from the ctenophore species *Mnemiopsis leidyi*[8,9] and the basal chordate *Branchiostoma lanceolatum*[5] have been cloned and expressed in vitro, revealing unique properties including variable sensitivity to the amino acid neurotransmitters glutamate and glycine. An interesting hypothesis that emerged from this and other comparative work, leveraging functional and structural observations of vertebrate and non-vertebrate iGluRs with bound ligands (reviewed in refs. [3,10]), is that only a small set of amino acids within the ligand binding domain (LBD) determine ligand specificity[8,11]. The most prominent ligands in these discussions are glutamate and glycine, and predictions of ligand specificity based on the noted set of residues in the LBD suggest that frequent and independent evolutionary switches in ligand

[1]Department of Biology, University of Toronto Mississauga, Mississauga, ON, L5L 1C6, Canada. [2]Department of Biochemistry and Biomedical Sciences, McMaster University, Hamilton, ON, L8S 4K1, Canada. [3]Sechenov Institute of Evolutionary Physiology & Biochemistry, Russian Academy of Sciences, Saint Petersburg, 194223, Russia. [4]School of Biology, University of Southampton, Southampton, UK. [5]Institute for Life Sciences, University of Southampton, Southampton, UK. [6]Clinical and Experimental Sciences, Faculty of Medicine, University of Southampton, Southampton, UK. [7]Integrated Program in Neuroscience, McGill University, Montreal, QC, H3G 0B1, Canada. [8]Department of Pharmacology & Therapeutics, McGill University, Montreal, QC, H3G 0B1, Canada. [9]These authors contributed equally: Boris S. Zhorov, Luis A. Yanez-Guerra, Alessandra Aleotti, Chloe C. Koens. ✉e-mail: adriano.senatore@utoronto.ca

specificity have occurred within each major class of metazoan iGluRs[11]. Although not yet experimentally validated, the relatively few amino acids that are presumably crucial for determining ligand specificity might explain why switches in specificity occurred so frequently[10,11]. Another notable feature of the characterized epsilon receptors from *M. leidyi* and *B. belcheri* is that their macroscopic currents undergo rectification at depolarized voltages, a hallmark feature of AMPA and kainate receptors attributable to voltage-dependent block by endogenous cytoplasmic polyamines[12]. Nonetheless, whether epsilon receptors are indeed modulated by polyamines, in a manner that is homologous to AKDF receptors, has not been rigorously examined.

An interesting group of animals in the context of metazoan iGluR evolution are the early diverging placozoans, which possess AKDF and epsilon receptors but lack NMDA receptors. These animals are millimetre-sized marine invertebrates that lack synapses and a nervous system but can coordinate their cells for directed locomotive behaviors including positive and negative chemotaxis[13–16], gravitaxis[17], and thermotaxis[18]. They also elicit behavioral responses to applied substances including glycine, glutamate, GABA, applied protons, and synthetic peptides that mimic endogenously encoded regulatory peptides[15,16,19,20]. Presumably, these soluble ligands are detected on cell surfaces leading to downstream cytoplasmic responses, for example contraction or altered ciliary beating that drives placozoan locomotion. Accordingly, in addition to AKDF and epsilon receptors, placozoans express a rich set of genes associated with neural signaling and detection of extracellular stimuli, including voltage-gated sodium, calcium, and potassium channels, mechanically gated channels such as TRP and piezo, other ligand-gated channels including P2X and Degenerin/Epithelial $Na^+$ channels (Deg/ENaCs), and metabotropic G protein coupled receptors [21–25]. To date, several studies have leveraged the basal phylogenetic position of placozoans, and specifically the species *Trichoplax adhaerens*, to explore the functional properties and evolution of such genes, including voltage-gated calcium ($Ca_V$) channels[26–28], Deg/ENaCs[29,30], and even one AKDF iGluR[31]. However, this AKDF receptor was found to be highly atypical in being constitutively active, and no placozoans epsilon receptors have been subject to functional characterization.

In this study, we generated a species-guided phylogeny of eukaryotic iGluRs, alongside a focused analysis of iGluR homologues from the four placozoan species *T. adhaerens*, *Trichoplax* species H2, *Hoilungia hongkongnesis*, and *Cladtertia collaboinventa*. Our work provides some insights into iGluR phylogeny in eukaryotes, and uncovers three conserved subclades of placozoan epsilon receptors, promoting us to propose a new naming scheme for these receptor types in placozoans. Our analysis also delineated one epsilon receptor from the species *T. adhaerens*, named GluE1αA, as an interesting target for functional characterization. First, based on comparisons of amino acid sequences in the LBD region, we predict GluE1αA to have recently switched its ligand specificity from glutamate to glycine, making it useful for experimentally testing whether ligand specificity can be altered back to glutamate by changing key amino acids in the LBD. Second, GluE1αA bears an atypical serine residue in its Q/R/N site, a region in the pore that defines cation specificity and influences regulation by polyamines, raising questions as to how these properties manifest compared to AMPA/kainate receptors with glutamine in this position, or NMDA receptors that have an asparagine.

Through electrophysiological characterization of GluE1αA expressed in CHO-K1 cells, combined with mutation analysis and structural modeling, we demonstrate that ligand specificity can indeed be experimentally altered, where mutation of just three amino acids in the LBD completely switched ligand specificity from glycine to glutamate, reversing the presumed evolutionary switch that occurred naturally for this receptor. Furthermore, these same mutations altered the pharmacological properties of GluE1αA, establishing a nascent sensitivity to AMPA and significantly increasing sensitivity to the AMPA/kainate receptor blocker CNQX. We also demonstrate that the serine reside in the Q/R/N site of GluE1αA imparts diminished regulation by polyamines and decreased $Ca^{2+}$ permeation compared to the canonical Q/R/N residues of glutamine and

asparagine. Furthermore, a glutamine to serine mutation in the Q/R/N site of the human GluA2 receptor similarly diminished $Ca^{2+}$ permeation and polyamine regulation, suggesting the general structure of the pore, including the placement of Q/R/N residues within the ion selectivity filter, is conserved between AKDF and epsilon receptors. We also demonstrate that, for both GluE1αA and human GluA2, a conserved aspartate residue four amino acids downstream of the Q/R/N site is critical for regulation by polyamines, since its mutation to alanine abolished polyamine block of macroscopic currents recorded via both whole-cell patch clamp and excised patch.

## Results

### Eukaryotic phylogeny of iGluRs and a revised nomenclature of placozoan iGluRs

We performed phylogenetic analysis to explore the evolutionary relationships of placozoan iGluRs within a broader phylogenetic context of metazoans and non-metazoan eukaryotes (Fig. 1a and Supplementary Figs. 1 to 4). To overcome challenges from inferring phylogenetic relationships between sequences of a single gene family from a wide range of distantly related species, we performed gene tree to species tree reconciliation (see Materials and Methods). Such species-tree-aware inference leverages species relationships to better resolve uncertain branch positions in a gene tree, and discriminates between duplication (D) and speciation (S) events at each branch[32]. Moreover, reconciling the gene tree with the species tree allows to root the resulting tree based on the information of species relationships. According to our analysis, animal-specific iGluRs include a broad AKDF group, an epsilon group that is paralogous to the AKDF receptors, and a NMDA group, which is paralogous to both AKDF and epsilon receptors (Fig. 1a and Supplementary Fig. 1). Ancient duplication events appear to have given rise to AMPA, kainate, and phi receptors in a bilaterian ancestor of protostomes and deuterostomes. While AMPA and kainate receptors are well represented in both deuterostomes and protostome invertebrates, phi receptors are less common in protostomes with a single clade of lophotrochozoan receptors and a single receptor from the ecdysozoan/arthropod species *Calanus glacialis* (Supplementary Fig. 1). For comparison, a previous phylogenetic analysis did not identify phi receptors outside of deuterostomes[5], while a more recent analysis similarly identified a clade of phi receptors in lophotrochozoans[1]. Our additional identification of a single phi receptor from the arthropod *C. glacialis* thus supports deep ancestry of phi receptors in protostome invertebrates but extensive losses in ecdysozoans (Supplementary Fig. 2).

Our analysis also supports a common origin of delta iGluRs in a basal bilaterian ancestor, finding phylogenetic homologues in deuterostome and lophtrochozoans from the phyla Mollusca and Annelida, consistent with previous studies[1,5,33]. Notably, one of these studies also identified a single delta receptor from the cnidarian species *Nematostella vectensis* (GluAKDF1 with accession number v1g50912)[1], suggesting an even deeper phylogenetic origin. However, this same protein sequence (accession number XP_001633354.1 in our analysis) formed a strong and separate clade with an expanded set of cnidarian AKDF receptors in both our gene tree and our species-reconciled tree (Fig. 1a, Supplementary Figs. 1 and 3). The phylogeny also delineates two clades of placozoan AKDF receptors, both positioned between delta receptors and a superclade of bilaterian AKDF receptors separated into group I (containing AMPA, kainate, and phi receptors), and group II (comprised of uncharacterized receptors from molluscs, annelids, cephalochordates, and the hemichordate *Ptychodera flava*). The earliest branching clade of placozoan receptors includes a single receptor from the calcareous sponge *Sycon ciliatum*, consistent with a previous report[1].

With respect to epsilon type receptors, we corroborate their existence in protostome invertebrates[1] by identifying numerous homologues from the marine annelid *Capitella teleta*, contrasting earlier studies that found them only in deuterostomes[5,6]. Our trees also support, as previously suggested, a pre-bilaterian origin of type 1 NMDA receptors, present also in cnidarians, and a bilaterian origin of NMDA2 and NMDA3 receptors, emerging via duplication of an ancestral NMDA2/3-like receptor after the cnidarian/

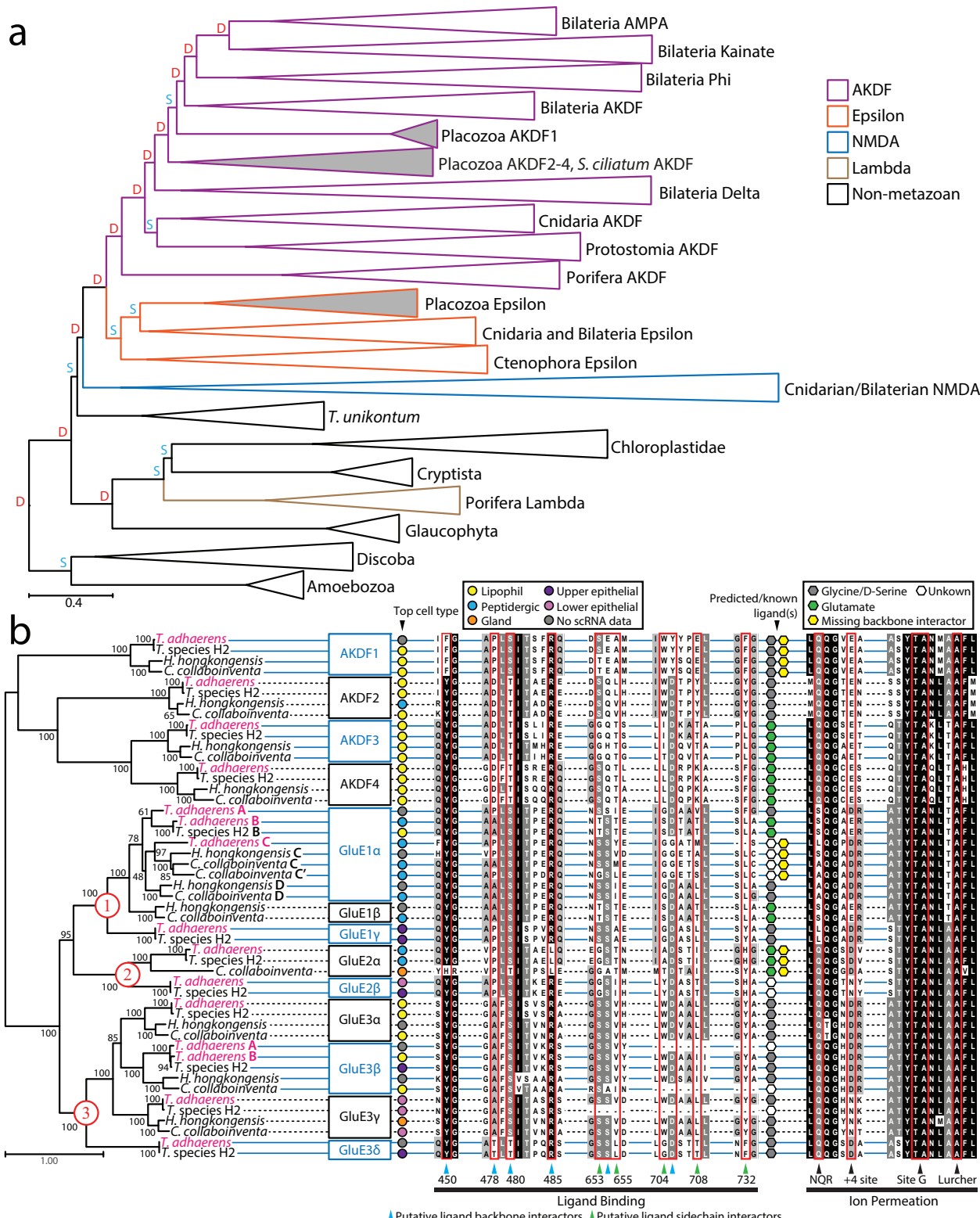

**Fig. 1 | Phylogeny and revised nomenclature of placozoan iGluRs. a** Species aware phylogenetic tree of identified iGluR proteins sequences from select eukaryotes and metazoans. The letters D (red) and S (cyan) denote predicted duplication and speciation and events, respectively, predicted by the GeneRax software[32]. **b** Maximum likelihood phylogenetic tree of placozoan iGluRs reveals invariable AKDF orthologues and a diversity of epsilon receptors falling within three major clades (1 to 3 labeled in red font). The black numbers on the nodes of the tree indicate ultrafast bootstrap values for 1000 replicates. Top cell type expression based from single cell

RNA-Seq data[25], predicted ligand selectivity for glycine/D-serine vs. glutamate are indicated by symbols to the right of the tree according to the corresponding legends, and aligned amino acids sequences associated with ligand binding and ion permeation are indicated below the alignments. Horizontal solid blue and dashed black lines running behind the aligned residues are used to demark different clades of placozoan iGluRs according to our proposed nomenclature indicated on the right of the phylogenetic tree.

bilaterian divergence[1,5,6] (Supplementary Fig. 1). Also consistent with previous studies, we find that the sponge-specific lambda receptors cluster outside of metazoan iGluRs[1,6,7]. However, in these previous studies the poriferan iGluRs were exclusively associated with sequences from the plant *Arabidopsis thaliana*, while our comprehensive analysis identified several additional sequences from Chloroplastida and Cryptista within the same lambda clade (Supplementary Figs. 1 and 2). This association is also apparent in a clustering analysis where the lambda receptors associate with these non-metazoan receptors and not metazoan iGluRs (Supplementary Fig. 4). Notably, while we identified lambda receptors in the gene data for species within the sponge clades Calcarea and Homoscleromorpha, we did not find any in the data for the three demosponge species included in our analysis (Supplementary Table 1). Two possible explanations for these various observations are that: 1) lambda receptors underwent strong sequence divergence in Porifera causing them to incorrectly associate with non-metazoan receptors, followed by gene loss in select clades within Porifera, or 2) as previously suggested[1], lambda receptors were passed from the Diaphoretickes lineage to a sub-clade of sponges via lateral gene transfer.

Although most clades in our species aware tree correspond with those in our iGluR gene tree, several clades differ between the two reflecting alterations incurred during species reconciliation (Supplementary Fig. 3). One of these is a clade of placozoan AKDF type 1 receptors, that in the gene tree associates with a set of deuterostome receptors, while in the reconciled tree forms a sister relationship with bilaterian group I and II AKDF receptors (Fig. 1a and Supplementary Fig. 3). Another difference is apparent for a clade of protostome AKDF receptors, which have an orthologous relationship with a set of cnidarian AKDF receptors in the reconciled tree but are included with the bilaterian Delta receptors in the gene tree. A third difference is apparent for a clade of receptors from the distant unicellular lineages of Amoebozoa and Discoba, which have a paralogous relationship with NMDA receptors in the gene tree but are sister to all other eukaryotic iGluRs in the reconciled tree (Fig. 1a and Supplementary Fig. 3). Lastly, there is a slight difference in the position of lambda receptors from sponges (Porifera), although in both the species and gene tree, they fall outside of metazoan iGluRs instead associating with receptors from species within Diaphoretickes.

Next, we re-examined the phylogeny of placozoan iGluRs using a more focused, manual approach for sequence identification. Specifically, we used BLAST and manual annotation to comprehensively identify iGluR sequences in the gene data available for the four placozoan species: *Trichoplax adhaerens*[21-23], *Trichoplax* species H2[23], *Hoilungia hongkongensis*[24], and *Cladtertia collaboinventa*[25], and used these sequences to generate a maximum-likelihood protein phylogeny (see Materials and Methods). The resulting tree revealed one-to-one orthology of AKDF receptors 1 to 4, contrasting the epsilon receptors with instances of lineage-specific duplication or loss producing more diversity between the four placozoan species (Fig. 1b). The comprehensive nature of our phylogenetic analysis, which resolves three major clades of placozoan Epsilon receptors, prompts a new nomenclature scheme based on phylogenetic groupings. Specifically, while AKDF1 to AKDF4 receptors retain their previous nomenclature, we suggest naming the three clades of placozoan epsilon receptors GluE1 to GluE3. *T. adhaerens* possesses four GluE1 paralogues, three within a strongly supported clade comprised of homologues from all other species (i.e., GluE1α subtypes A to C), and a fourth, also found in *Trichoplax* species H2 (GluE1γ), forming a sister relationship with other GluE1 sequences. Several nodes within the GluE1 clade are poorly supported, making it difficult to infer internal phylogenetic relationships. However, it is notable that *H. hongkongensis* and *C. collaboinventa*, which are more phylogenetically related to each other[25], uniquely share GluE1β receptor orthologues, while *T. adhaerens* and *Trichoplax* species H2, also more closely related, uniquely possess GluE1γ receptors. In the GluE2 clade, there is one-to-one orthology of GluE2α and GluE2β receptors between *T. adhaerens* and *Trichoplax* species H2, and a single GluE2α orthologue from *C. collaboinventa* but none identified for *H. hongkongnesis*. More cross-species conservation is apparent in the GluE3 clade, with one-to-one orthologous relationships for GluE3α,

GluE3β, and GluE3γ receptors. The exceptions are GluE3β, which uniquely duplicated in *T. adhaerens* giving rise to GluE3βA and GluE3βB, and GluE3δ, which we only found in the gene data from *T. adhaerens* and *Trichoplax* species H2. Based on the deep and strongly supported nodes of the three epsilon clades within Placozoa, we suggest this proposed nomenclature will withstand incorporation of new sequences should they become available. In Supplementary Table 2, we summarize this naming scheme and provide corresponding accession numbers and previous names of all relevant receptors.

Next, we explored whether the phylogenetic relationships of placozoan iGluRs are also reflected by their mRNA expression levels in homologous cell types defined for placozoans. To do so, we mined the available single cell RNA-Seq data[25] to determine the top cellular expression of each receptor, revealing that the orthologous AKDF1 to AKDF4 receptors are most expressed in digestive lipophil cells (Fig. 1b), except for AKDF2 from *H. hongkongensis*, which is most abundant in neuron-like peptidergic cells despite also having strong expression in lipophil cells. In contrast, most of the GluE1 epsilon receptors show enriched expression in peptidergic cells, with a notable exception being GluE1γ from *T. adhaerens* and *Trichoplax* species H2, which are enriched in upper/dorsal epithelial cells. More diversity is apparent within the GluE2 and GluE3 clades, though GluE3α and GluE3β show a general enriched expression in lipophil cells, and GluE3γ in lower/ventral epithelial cells.

We also predicted the ligand specificities of the various placozoan iGluRs based on residues in aligned sequences that form key contacts with amino acid ligands in other receptors and are proposed to be deterministic for ligand specificity (reviewed in ref. [11]). Specifically, while residues at positions 450, 478, 480, 485, 654, and 705 interact with backbone atoms found in all amino acid ligands (numbered according to alignment with the mature rat GluA2 subunit with UniProt accession number P19491; Supplementary Data 1), residues 653, 655, 704, 708, and 732 form contacts with ligand side chain atoms and hence play key roles in defining specificity. Thus, the presence of glycine, serine, or threonine at position 653, in conjunction with threonine at position 655 and tyrosine at position 732 is associated with glutamate specificity, while serine at position 653, along with a hydrophobic residue at position 655 (valine, leucine, alanine, isoleucine, or proline) and phenylalanine at position 732, is associated with glycine/D-serine specificity[11]. Based on these contacts, placozoan AKDF1 and AKDF2 receptors are predicted to bind glycine, while AKDF3 and AKDF4 are predicted to bind glutamate (Fig. 1b). Notably however, the AKDF1 receptors lack an aspartate 750 ($D_{705}$) residue, which engages in electrostatic interactions with the ligand alpha amino group and is important for ligand binding. Recently, this natural variation in sequence was shown to cause the *T. adhaerens* AKDF1 receptor to conduct constitutive leak currents that are blocked by a broad range of ligands, most pronounced for glycine[31]. A subset of GluE1α epsilon receptors are similarly missing the $D_{705}$ residue and have variable sequences at other key sites that prevents prediction of their ligand specificities (Fig. 1b). The GluE2α orthologues also lack key ligand backbone interacting residues, including arginine 485 ($R_{485}$) which forms key electrostatic contacts with the ligand alpha carboxyl group but are predicted to bind glutamate. The consequences of these missing backbone-interacting residues in GluE1α and GluE2α receptors are unknown. Among epsilon receptors with predictable ligand specificity profiles, all GluE3 LBDs are predicted to bind glycine, while GluE2 receptors are predicted to bind glutamate. Instead, extensive ligand switching is apparent in the GluE1 LBD, with GluE1γ predicted to bind glycine, GluE1β to bind glutamate, and GluE1α to bind either glycine or glutamate. Considering GluE1 receptors from *T. adhaerens*, it is notable that GluE1αA is predicted to bind glycine, while its close paralogue GluE1αB, and its orthologue from *Trichoplax* species H2, are predicted to bind glutamate. Based on our phylogeny, these are the two most closely related receptors with differing predicted ligand specificity, sharing 62% amino acid sequence identity and 81% similarity in the LBD (i.e., amino acids $T_{430}$ to $T_{543}$ and $L_{692}$ to $F_{816}$ of GluE1αA, and $N_{15}$ to $R_{128}$ and $L_{277}$ to $F_{401}$ of GluE1αB). Thus, it seems plausible that a recent evolutionary switch occurred, where either GluE1αA evolved glycine

specificity from a glutamate-selective ancestor, or GluE1αB evolved glutamate specificity from a glycine-selective ancestor.

We examined sequences in the pore-loop (P-loop) region located between the transmembrane helices M1 and M3 of each subunit, as well as the M3 helices which together with the P-loop govern ion permeation and selectivity. Most placozoan iGluRs possess a glutamine in the Q/R/N site, a key locus that defines monovalent and divalent cation selectivity[3,34], in this way resembling AMPA and kainate receptors (Fig. 1b). GluE1 subunits are the exception having non-canonical residues of serine, alanine, or leucine at this position. All placozoan iGluR subunits, except GluE2β and GluE3γ, possess negatively charged aspartate or glutamate residues four amino acids downstream of the Q/R/N site, which together with Q/R/N residues contribute to polyamine binding in the pore and voltage-dependent block of ion permeation (reviewed in ref. 12). Lastly, some sequence divergence is apparent in the M3 helix bearing the SYTANLAAF motif, within a typically highly conserved structure which in tetrameric iGluRs contributes to the gate that opens during channel activation[35]. Nonetheless, complete conservation is evident for alanine residues in the lurcher position (Fig. 1b), a site which when mutated produces constitutively open channels[36], and alanine-threonine residues in site G, recently shown to be involved in $Ca^{2+}$ permeation[37].

### The *Trichoplax* GluE1αA receptor has a broad ligand specificity and moderately fast recovery from desensitization

Since the available transcripts for the *T. adhaerens* iGluRs were algorithmically assembled from RNA-Seq reads, we sought to verify their coding sequences by amplifying them in full-length (i.e., start to stop codon) from a whole animal cDNA library via PCR, followed by cloning into the mammalian expression vector pIRES2-EGFP and sequencing in triplicate. This led to the successful cloning and verification of 10 iGluRs, whose sequences we submitted to GenBank (see Materials and Methods). When cloning the various iGluRs, we included a consensus mammalian Kozak sequence of GCCGCCACC upstream of the start codon allowing protein expression of the channels in transfected Chinese Hamster Ovary (CHO)-K1 cells for whole-cell patch-clamp electrophysiology and functional characterization. In preliminary experiments, we were able to observe ligand-gated current for several AKDF and epsilon receptors, including robust currents for the GluE1αA receptor expressed on its own and hence forming a functional homomeric channel. We decided to conduct a comprehensive functional analysis on this latter receptor, considering its putative evolutionary switch in ligand specificity relative to the GluE1αB receptors from *T. adhaerens* and *Trichoplax* species H2, and its atypical serine residue in the Q/R/N site. As predicted from sequences in its LBD region (Fig. 1b), GluE1αA is activated by glycine producing robust currents in CHO-K1 cells that are also activated by D-serine, L-serine, and alanine, and weakly by valine (Fig. 2a). We observed no responses to glutamate and aspartate (Fig. 2a), nor for arginine, threonine, methionine, or betaine tested on their own at 10 mM, nor for histidine, lysine, isoleucine, leucine, and phenylalanine tested as a single mix each at 3 mM. Dose-response curve analysis of activating ligands revealed strongest sensitivity to alanine, followed by glycine, L-serine, D-serine, all with $EC_{50}$ values in the sub-millimolar range (i.e., L-alanine = $0.049 \pm 0.011$ mM, glycine = $0.11 \pm 0.02$ mM, L-serine = $0.13 \pm 0.02$ mM, and D-serine = $0.19 \pm 0.04$ mM), followed by valine with an $EC_{50}$ of $1.51 \pm 0.24$ mM (Fig. 2b and c, Supplementary Fig. 5). Altogether, GluE1αA most resembles the vertebrate NMDA receptor subunit GluN1 and the *Mnemiopsis leidyi* (ctenophore) epsilon subunit ML032222a in its ligand specificity, both also activated by glycine, alanine, and serine[3,8]. In contrast, the epsilon iGluR from the basal chordate *Branchiostoma lanceolatum*, GluE1, is not sensitive to alanine, serine, or glutamate, despite being activated by glycine[5].

Perfusion of a 2 second pulse of 3 mM glycine, followed by washes of increasing duration followed by a 2 second pulse of 3 mM glycine, revealed fast monophasic recovery from desensitization for GluE1αA with a time constant of $184 \pm 45$ milliseconds (Fig. 2d and e). The *T. adhaerens* receptor is thus much faster in its recovery compared to the epsilon receptor from the

lancelet *B. lanceolatum* with a time constant of 10.8 seconds[5], and the ctenophore *M. leidyi* with a time constant of 81 seconds, the latter attributed to a unique interdomain salt bridge which contributes to a very slow recovery from desensitisation[8,9].

### Mutation of key residues in the ligand binding domain switches ligand specificity from glycine to glutamate

As noted, a model has been put forward that defines iGluR ligand specificity based on a set of key amino acids in the LBD of iGluRs[11]. This prediction scheme proved correct for GluE1αA (Figs. 1b and 3a), which we found to be insensitive to glutamate but activated by glycine and D-/L-serine (Fig. 3b and c). To the best of our knowledge, whether these few positions are indeed solely responsible for ligand specificity, and moreover, can be altered to switch specificity for natural ligands, has not been experimentally tested. Referring to this model, we mutated the serine and isoleucine residues at positions 653 and 655 of GluE1αA to glycine and threonine, respectively, to convert its predicted ligand specificity from glycine to glutamate (named the m2 variant, with the mutations $S_{653}G$ and $I_{655}T$) (Fig. 3a). We also created a triple mutant, with an additional phenylalanine to tyrosine mutation at position 732 ($F_{732}Y$; m3 variant), to make the *T. adhaerens* epsilon receptor resemble GluN2A and GluA2 receptors from human, reasoning that this mutation might enhance glutamate sensitivity. We also generated a single $F_{732}Y$ mutant (m1 variant), to assess the contribution of this amino acid to ligand specificity in isolation.

Application of various activating ligands at 10 mM concentrations and normalizing peak currents to either glycine (wildtype and m1), or glutamate (m2 and m3), revealed that the m2 variant of GluE1αA became sensitive to glutamate, while becoming proportionately less sensitive to glycine, L-serine, D-serine, alanine, and valine (Fig. 3b and c). The triple mutant (m3), bearing the additional $F_{732}Y$ mutation, almost completely lost sensitivity to the tested non-glutamate ligands, in lieu of robust glutamate-activated currents. On its own, the $F_{732}Y$ mutation did not diminish the relative sensitivity to glycine nor did it cause glutamate sensitivity, instead causing decreased sensitivity to D-serine, increased sensitivity to L-serine and alanine, and completely attenuated sensitivity to valine. Dose response analysis revealed that the single $F_{732}Y$ mutant had decreased sensitivity to glycine compared to wildtype (i.e., glycine $EC_{50} = 0.11 \pm 0.02$ mM for wildtype vs. $0.30 \pm 0.04$ for m1), while it increased sensitivity to glutamate when created alongside the $S_{653}G$ and $I_{655}T$ mutations (i.e., glutamate $EC_{50} = 1.81 \pm 0.20$ mM for m2 vs. $0.61 \pm 0.06$ for m3) (Fig. 3d).

To gain insights into why these amino acid changes resulted in a ligand specificity switch, we predicted the structure of GluE1αA with AlphaFold[38,39] and conducted docking of glycine and glutamate to the ligand binding pocket of the wildtype and m3 receptor variants. Interestingly, the Alpha-Fold 3 predicted tetrameric structure of the *T. adhaerens* epsilon receptor (Fig. 3e) resembles the resolved structures of vertebrate homotetrameric GluD1 and GluD2 delta receptors, with the N-terminal domain (NTDs) and ligand binding domains (LBDs) forming a dimer of dimers arrangement, as opposed to the NTD-LBD swapped domains apparent for AMPA and kainate receptors (reviewed in ref. 4). We also generated AlphaFold models of the isolated LBD and transmembrane domain (TMD) of wildtype and m3 variant of GluE1αA and docked glycine or glutamate with a starting position that mimicked the location of D-serine bound to the LBD of rat GluD2, determined via cryo-EM (PDB ID 2v3u[40]). Intensive Monte Carlo energy-minimizations yielded complexes shown in Fig. 3f to I and Supplementary Fig. 6. The energy scores of ligand-receptor interactions are provided in Supplementary Table 3a to c. Glycine binding to the wildtype channel is stabilized mainly by salt bridges with residues $D_{705}$ and $R_{485}$ and non-bonded attraction to $K_{728}$ and $Y_{450}$ (Fig. 3f; Supplementary Fig. 6a; Supplementary Table 3b). Of note, the total energy scores of glycine bound to the wildtype and m3 receptor variants reflect experimental observations, with the m3 variant having a higher energy score of $-2.51$ kcal/mol (i.e., weaker) compared to $-5.92$ kcal/mol for wildtype (Supplementary Table 3b), mainly due to weaker salt bridges and repulsive

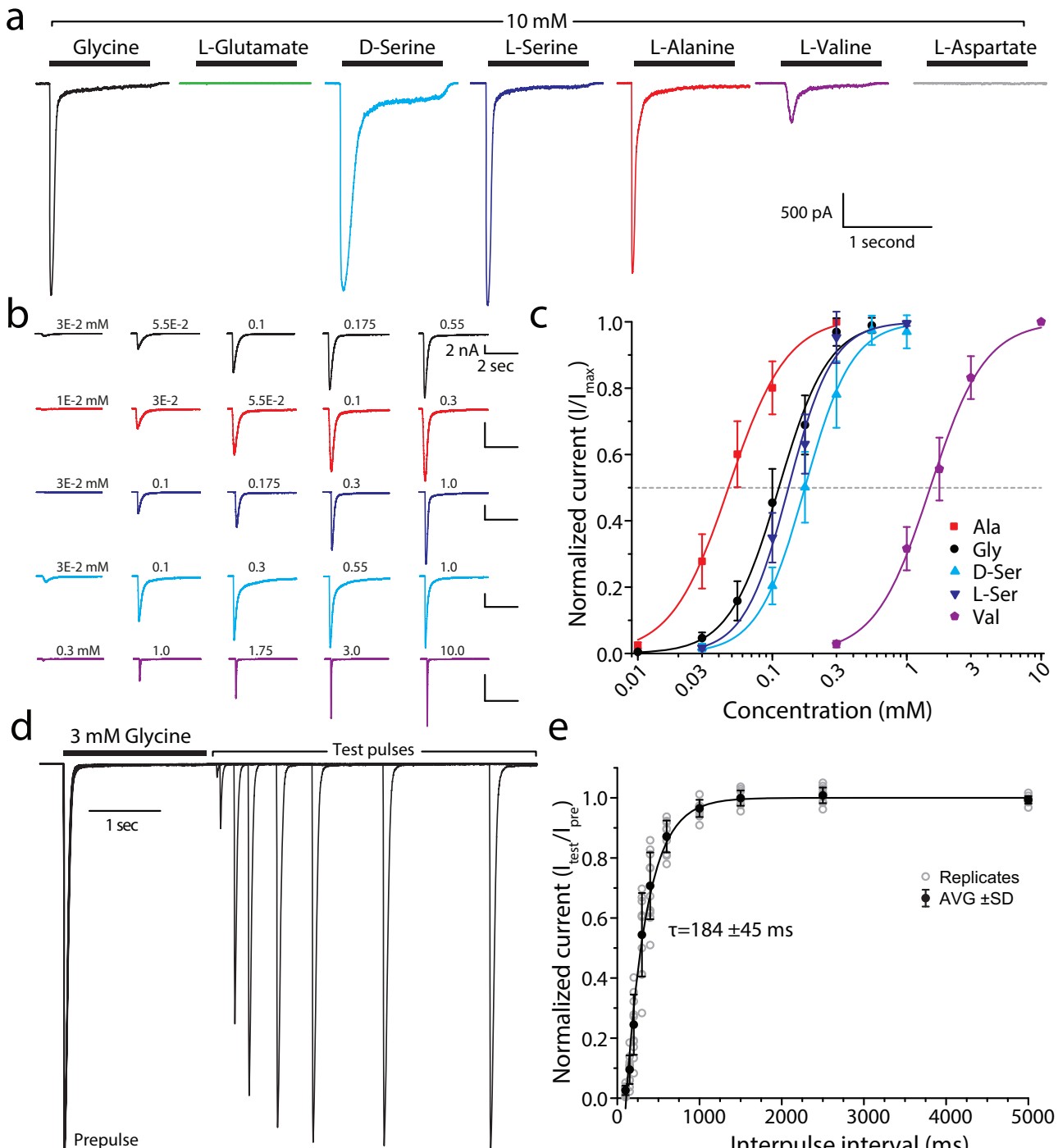

**Fig. 2 | In vitro currents of the *T. adhaerens* GluE1αA epsilon receptor. a** Sample whole-cell currents of the *T. adhaerens* GluE1αA receptor expressed in CHO-K1 cells upon extracellular perfusion of 10 mM glycine, glutamate, L-serine, D-serine, alanine, valine, and aspartate. **b** Sample recordings of GluE1αA receptor macroscopic currents elicited by increasing concentrations of amino acid ligands (top to bottom: glycine in *black*, alanine in *red*, L-serine in *navy blue*, D-serine in *light blue*, and valine in *purple*). **c** Dose response curves showing greatest sensitivity to alanine, and lowest sensitivity to valine. The EC$_{50}$ values for all ligands were statistically different from glycine as shown by two-sample T-tests ($p < 0.05$). **d** Sample paired pulse currents of GluE1αA elicited by 3 mM glycine applied for 2 seconds (to desensitize the channels), followed by a wash step of increasing duration then 2 second test pulse. **e** Plot of average recovery from desensitization of GluE1αA after prolonged application of 3 mM glycine, fitted to a single exponential with a τ value of 184 ± 45 ms ($n = 10$).

contributions from the mutated Y$_{732}$ residue. Furthermore, Y$_{732}$ is predicted to hydrogen bond with D$_{705}$, shifting the latter away from the ligand and hence contributing to the lower affinity for glycine in the m3 LBD. The energy score of glutamate in the wildtype channel was positive (unfavorable) mainly due to repulsion from D$_{705}$, S$_{480}$, and E$_{656}$, while a salt bridge between the glutamate ligand and R$_{485}$ provides a strong attractive contribution. The interaction energy score of glutamate with the mutant channel was negative mainly due to attractive contributions from R$_{450}$, K$_{728}$, as well as the mutated residues G$_{653}$, T$_{655}$. Besides indicated residues, which provide major attractive or repulsive contributions to ligand-channel interactions, there are other residues that contribute energies with absolute value over 0.1 kcal/mol (Supplementary Table 3c). For

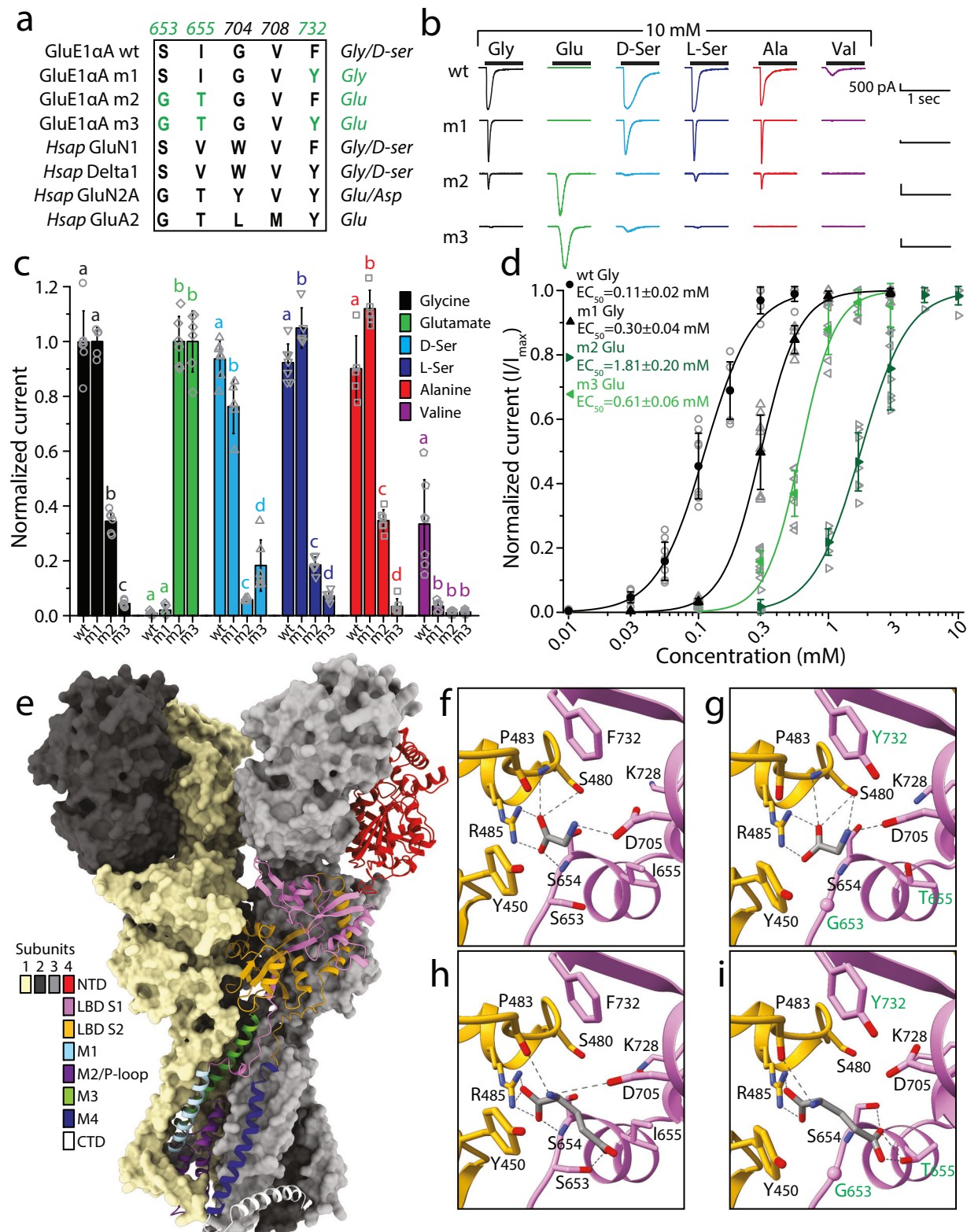

these analyses, it should be emphasized that the estimated interaction energy scores between ligands and the LBD were weak, indicating that these are sensitive to imprecision of the used force field. Indeed, although our computations are consistent with the trend of ligand-channel energy changes, the predicted energy scores should not be considered as measures of ligand affinity.

**Mutations that alter ligand specificity also affect sensitivity to vertebrate iGluR agonists and the AMPA/kainate blocker CNQX**

Since pharmacological agonists of mammalian iGluRs interact with residues in the LBD, with some structural overlap with binding sites for natural ligands (reviewed in ref. 4), we asked whether the three amino acid changes that converted the ligand specificity of GluE1αA from glycine to glutamate

**Fig. 3 | Mutation of key residues in the ligand binding domain of GluE1αA narrows its ligand-selectivity by switching to glutamate. a** Multiple sequence alignment of deterministic residues that interact with ligand amino acid side chains and mediate ligand specificity in mammalian NMDA, AMPA, and kainate receptors (numbers correspond to amino acids positions in rat GluA2). The residues in green were mutated in the wildtype (wt) GluE1αA receptor to resemble the glutamate-activated GluN2A and GluA2 receptors from human (the single, double and triple mutant variants are referred to as m1, m2 and m3 respectively). **b** Sample whole-cell currents of wt, m1, m2 and m3 variants of GluE1αA elicited by application of different amino acids at 10 mM. **c** Plot of average normalized peak inward currents of wt, m1, m2 and m3 variants of GluE1αA in response to different amino acid ligands at 10 mM ($n$ = 5-7). Letters above the bars denote statistically significant differences for ligand type based on post hoc Tukey tests ($p < 0.05$) after One Way ANOVAs (glycine: F = 290.12, $p = 1.11\text{E}{-}16$; glutamate: F = 336.17, $p = 1.11\text{E}{-}16$; D-serine: F = 209.73, $p = 2.78\text{E}{-}15$; L-serine: F = 534.65, $p = 0$; alanine: F = 279.39, $p = 2.78\text{E}{-}15$; valine: F = 20.76, $p = 2.35\text{E}{-}6$). Specifically, means/bars with the same letters above them are not statistically different, while those with different letters are different based on Tukey tests. **d** Dose response curves of wt and m1 variants GluE1αA normalized current responses to glycine ($n$ = 8), and of the m2 and m3 variants to glutamate ($n$ = 6). One Way ANOVA confirmed significant differences among the $EC_{50}$ values (F = 343.15, $p = 0$), and Tukey post hoc tests revealed that all three were different from each other ($p < 0.01$). **e** AlphaFold 3-predicted structure of the GluE1αA tetramer. Three of the four subunits are shown as surfaces colored pale goldenrod, black, and gray, while the fourth subunit is depicted as a ribbon structure with different structural regions colored differently as indicated by the legend. NTD: N-terminal domain; LBD S1: ligand binding domain segment 1; LBD S2: ligand binding domain segment 2; M1 to M4: transmembrane helices 1 to 4; P-loop: pore-loop; CTD: C-terminal domain. **f** Homology modeling and docking of glycine in the putative ligand binding pocket of the wildtype GluE1αA receptor. **g** Homology modeling and docking of glycine in triple mutant (m3) variant of GluE1αA. **h** Homology modeling and docking of glutamate in the wt GluE1αA receptor. **i** Homology modeling and docking of glutamate in triple mutant (m3) variant of GluE1αA. In **f** to **i**, predicted hydrogen bonds are depicted by dashed grey lines.

also altered sensitivity to AMPA, kainate, NMDA, and quisqualate. Perfusing different extracellular solutions over recorded cells expressing the wildtype receptor revealed robust currents elicited by 1 mM glycine, but insensitivity to 1 mM AMPA, kainate, and NMDA (Fig. 4a, b). Similarly, the m2 variant of GluE1αA produced robust responses to 1 mM glutamate but was insensitive to the tested agonists. In contrast, the m3 variant exhibited moderate responses to 1 mM AMPA, with the drug causing slow desensitizing currents with peak amplitudes roughly 36% of those elicited by 1 mM glutamate (Fig. 4a, b). We also tested the competitive AMPA/kainate receptor blocker CNQX on the wildtype and m3 variants of GluE1αA, uncovering a low affinity block of glycine-activated currents for the wildtype receptor with an $IC_{50}$ of 50 ± 17 μM. Interestingly, the m3 variant exhibited a biphasic response, with 1 μM CNQX causing a roughly 20% increase in mean peak current amplitude ($EC_{50}$ = 1.0 ± 0.1 μM for a biphasic curve fit over the data), followed by block of glutamate-activated currents with concentrations >1 μM ($IC_{50}$ = 6.7 ± 0.9 μM for a monophasic curve fit over the blocked current, and 7.7 ± 0.5 μM for a biphasic curve fit; Fig. 4c to e). Altogether, mutations that alter ligand specificity can also alter sensitivity to pharmacological compounds that associate with the ligand binding site of iGluRs.

## A unique serine residue in the Q/R/N site decreases voltage-dependent regulation by polyamines

Interestingly, variable sequences in the Q/R/N site of placozoan group 1 epsilon receptors (Fig. 1b) suggests dynamic evolution of ion selectivity and polyamine regulation in this clade. In mammalian AMPA, kainate, and NMDA receptors, the Q/R/N site forms the narrowest part of the ion permeation pathway and is a locus for governing ion permeation and selectivity. Heteromeric AMPA and kainate receptors for example, which encode glutamine in the Q/R/N site at the mRNA level, exhibit different $Ca^{2+}$ permeation properties depending on subunit composition, where A to I mRNA editing of select subunits converts the Q/R/N glutamine to arginine, rendering channels that are impermeable to $Ca^{2+}$ (reviewed in refs. [3,34]). NMDA receptors do not exhibit A to I editing, but an asparagine to lysine mutation in the otherwise invariable Q/R/N site of the NMDA2A subunit is associated with severe developmental delay and early-onset epileptic encephalopathy, and in vitro this mutation causes severely diminished $Ca^{2+}$ permeation and diminished voltage-dependent $Mg^{2+}$ block of conducted currents[41]. Interestingly, while all placozoan AKDF receptors, and all clade 2 and 3 epsilon receptors have glutamine resides in the Q/R/N site, thus resembling mammalian AMPA and kainate receptors, all GluE1 subunits possess divergent residues of either alanine, leucine, or serine (Fig. 1b). Hence, to determine if the non-canonical serine residue in the GluE1αA Q/R/N site impacts ion permeation and/or selectivity, as well as polyamine regulation, we characterized its ion permeation properties in more detail. First, we conducted bi-ionic reversal potential experiments with an invariable internal solution containing 150 mM $Na^+$ and used perfusion to switch

the extracellular solution from equimolar $Na^+$ to $Li^+$, $K^+$, and $Cs^+$ (Fig. 5a). Eliciting currents with 3 mM glycine while changing the holding voltage to between −100 and +80 mV, produced currents with reversal potentials (i.e., $E_{Rev}$ values) near zero mV for all bi-ionic conditions (Fig. 5b–e). For the wildtype receptor, converting differences in $E_{Rev}$ values for the different bi-ionic conditions to $pX^+/pNa^+$ permeability ratios (where X = Na, Li, K, or Cs) revealed a slight preference for $Na^+$ over $Li^+$ and $Cs^+$ ions, and a slight preference for $K^+$ over $Na^+$ (Fig. 5f; statistical comparisons provided in Supplementary Table 4a). Mutation of the serine Q/R/N residue to glutamine ($S_{642}Q$), to resemble other placozoan iGluRs and AMPA/kainate receptors, abrogated the marginal permeability differences of the wildtype receptor, with all $pX^+/pNa^+$ values being statistically indistinguishable from each other. Similarly, mutation to asparagine ($S_{642}N$), to resemble NMDA receptors, also abrogated selectivity preferences except for $Li^+$ which retained its marginally lower permeability compared to $Na^+$. We also mutated the serine Q/R/N to arginine, resulting in a lack of selectivity between $Na^+$ and $Li^+$, and slightly diminished permeability of $K^+$ and $Cs^+$ relative to $Na^+$ (Fig. 5f). Altogether, it appears as though the serine in the Q/R/N site of GluE1αA renders a channel that is largely non-selective between monovalent cations, being similar in this respect to the glutamine and asparagine residues found in AMPA, kainite, and NMDA receptors.

Despite only slight differences in monovalent ion selectivity, we did notice marked differences in the rectification of currents through changing voltage for the different channel variants. Specifically, while the wildtype receptor exhibited only slight rectification with an almost linear current-voltage (I-V) relationship, the $S_{642}N$, $S_{642}Q$, and $S_{642}R$ variants showed considerably increased current rectification near 0 mV (Fig. 5b to e). Converting the I-V plots to conductance-voltage (G-V) plots, which removes the effect of driving force and elucidates changes in conductance as a function of voltage, revealed a continued decline in conductance for the wildtype channel with increasing depolarization up to +20 mV, followed by a subsequent increase at more positive voltages (Fig. 5g). This voltage-dependent drop in conductance, also apparent for the in vitro expressed epsilon receptors from the cephalochordate *B. lanceolatum* and the ctenophore *M. leidyi*[5,8], is consistent with a mild voltage-dependent block of GluE1αA by polyamines. These compounds, namely spermine, spermidine, and putrescine, are endogenously expressed in the cytoplasm of all living cells[42], where they regulate various targets including AMPA, kainate, and delta receptors[12,43,44]. Indeed, previous studies have demonstrated that polyamine block depends on Q/R/N residues, as well as a conserved aspartate 4 amino acids downstream (i.e., the +4 site)[12]. These linear cationic molecules are thought to insert into the pore selectivity filter at depolarized voltages, facilitated by electrostatic interactions with Q/R/N and +4 aspartate residues, resulting in obstructed ion permeation[45].

Notably, although current rectification and a corresponding drop in conductance was apparent for the wildtype GluE1αA receptor, it became much more pronounced in the AMPA/kainate/NMDA-like variants

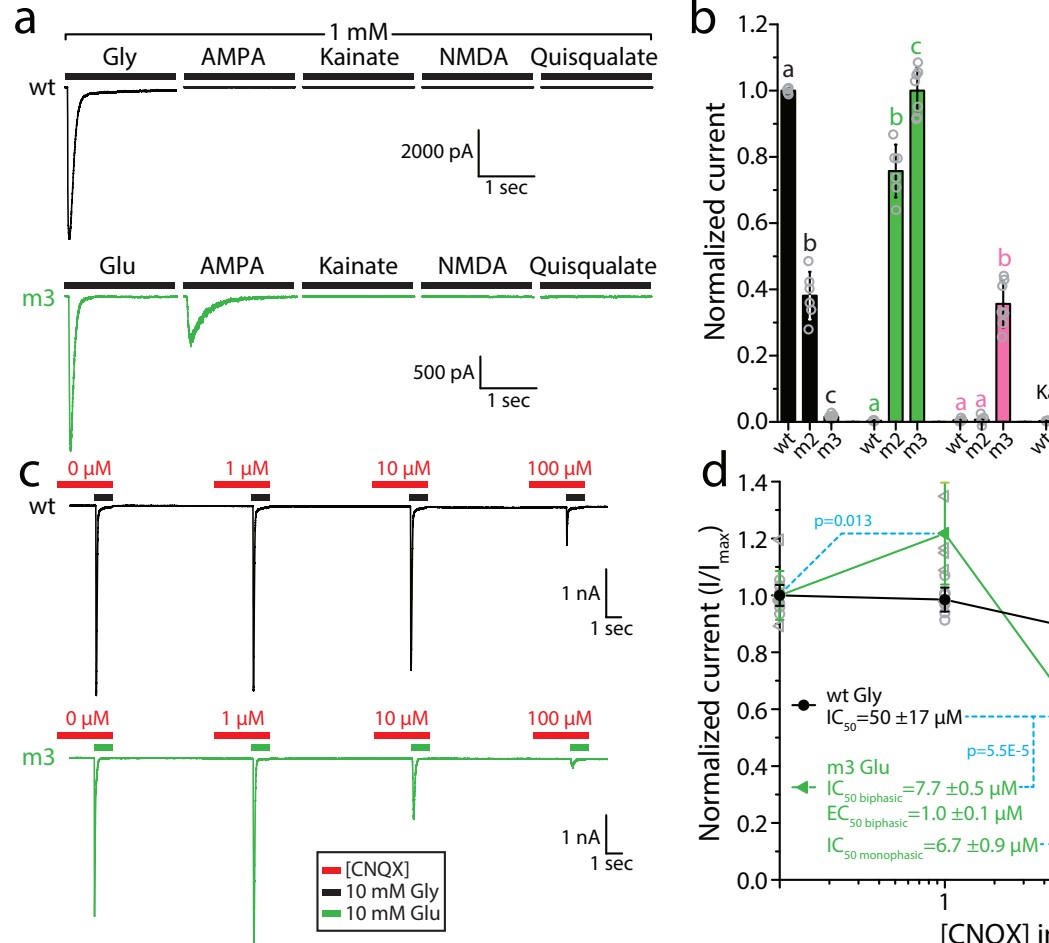

**Fig. 4 | GluE1αA mutations that alter ligand specificity also affect pharmacological sensitivity to CNQX and AMPA. a** Sample whole-cell recordings demonstrating that GluE1αA is not sensitive to 1 mM AMPA, kainate, NMDA, or quisqualate, while the m3 variant is sensitive to AMPA. **b** Bar plot of average normalized peak inward currents of wt, m2, and m3 variants of GluE1αA in response to different agonists at 1 mM ($n = 6$–7). Letters above the bars denote statistically significant differences for applied compounds based on post hoc Tukey tests ($p < 1E$−4) after One Way ANOVAs (glycine: F = 809.23, $p = 5.55E{-}16$; glutamate: F = 367.97, $p = 1.79E{-}13$; AMPA: F = 126.50, $p = 4.07E{-}10$). Specifically, means/bars with the same letters above them are not statistically different, while those with different letters are different based on Tukey tests. **c** Sample continuous GluE1αA wt and m3 currents elicited by 1 second perfusion of 10 mM glycine or glutamate with

increasing concentrations of 6-cyano-7-nitroquinoxaline-2,3-dione (CNQX; applied 2 seconds prior to ligands and during ligand application). The legend at the bottom indicates the duration of perfusion of glycine or glutamate ligands and CNQX. **d** CNQX dose response curves for the wildtype and m3 variants of GluE1αA normalized to peak current amplitude in the absence of CNQX ($n = 6$-9). $IC_{50}$ values for the wildtype receptor, derived using a standard monophasic dose response function, and $IC_{50}$ and $EC_{50}$ values for the m3 variant, derived using a biphasic dose response function[87], are indicated. The $p$-values indicated in cyan text are for two sample T-tests comparing normalized mean current amplitude of the GluE1αA m3 variant in the absence of CNQX vs. 1 μM CNQX, and of the mean $IC_{50}$ values of wildtype GluE1αA vs. the m3 variant.

bearing glutamine, asparagine, or arginine in the Q/R/N site (Fig. 5b–e, g and Supplementary Fig. 7). Indeed, the difference in normalized conductance values at 0 mV between the wildtype receptor and the $S_{642}Q$, $S_{642}N$, and $S_{642}R$ variants revealed negative changes ranging from $34.3 \pm 12.9\%$ to $45.3 \pm 9.8\%$, reflecting considerable increases in polyamine block of the variant channels (Fig. 5g inset). As noted earlier, most placozoan iGluRs, including GluE1αA, have a conserved acidic (i.e., aspartate or glutamate) residue in the +4 site (Fig. 1a), also apparent in most metazoan epsilon and AKDF receptors[5]. We thus mutated the +4 aspartate GluE1αA to alanine ($D_{646}A$), causing only marginal effects on monovalent ion selectivity, with a slight but statistically significant increase in $K^+$ over $Na^+$ permeability (Fig. 5f). However, this mutation resulted in completely linear I-V and G-V (Fig. 5b–e, g and Supplementary Fig. 7), and a positive change in normalized conductance at 0 mV relative to wildtype (i.e., $26.4 \pm 9.8\%$; Fig. 5g inset), indicating a pronounced loss of polyamine block. Similar observations were made for a double mutant bearing the $S_{642}Q$ mutation, which on its own increased polyamine block, paired with the

$D_{646}A$ mutation, pointing to a conserved and dominant function of the +4 residue in polyamine regulation.

To determine whether these various observations are relevant in a broader context, we conducted similar experiments on the human GluA2 receptor bearing either a wildtype glutamine in the Q/R/N site, or serine like GluE1αA. Interestingly, while the wildtype receptor exhibited characteristic rectifying currents and pronounced U-shaped G-V curves under all ionic conductions (activated by 3 mM glutamate), the serine variant had much more linear I-V curves and diminished drops in conductance with increasing depolarization (Fig. 5b–e, g and Supplementary Fig. 7). This mutation did not cause appreciable changes in current reversal potentials, although the strong rectification of the wildtype receptor prevented us from accurately determining $E_{Rev}$ values for making statistical comparisons. Nonetheless, it seems that placing a serine in the GluA2 Q/R/N site does not impose strong changes in monovalent ion selectivity, like GluE1αA. Lastly, comparing the difference in normalized conductance at 0 mV between the wildtype AMPA

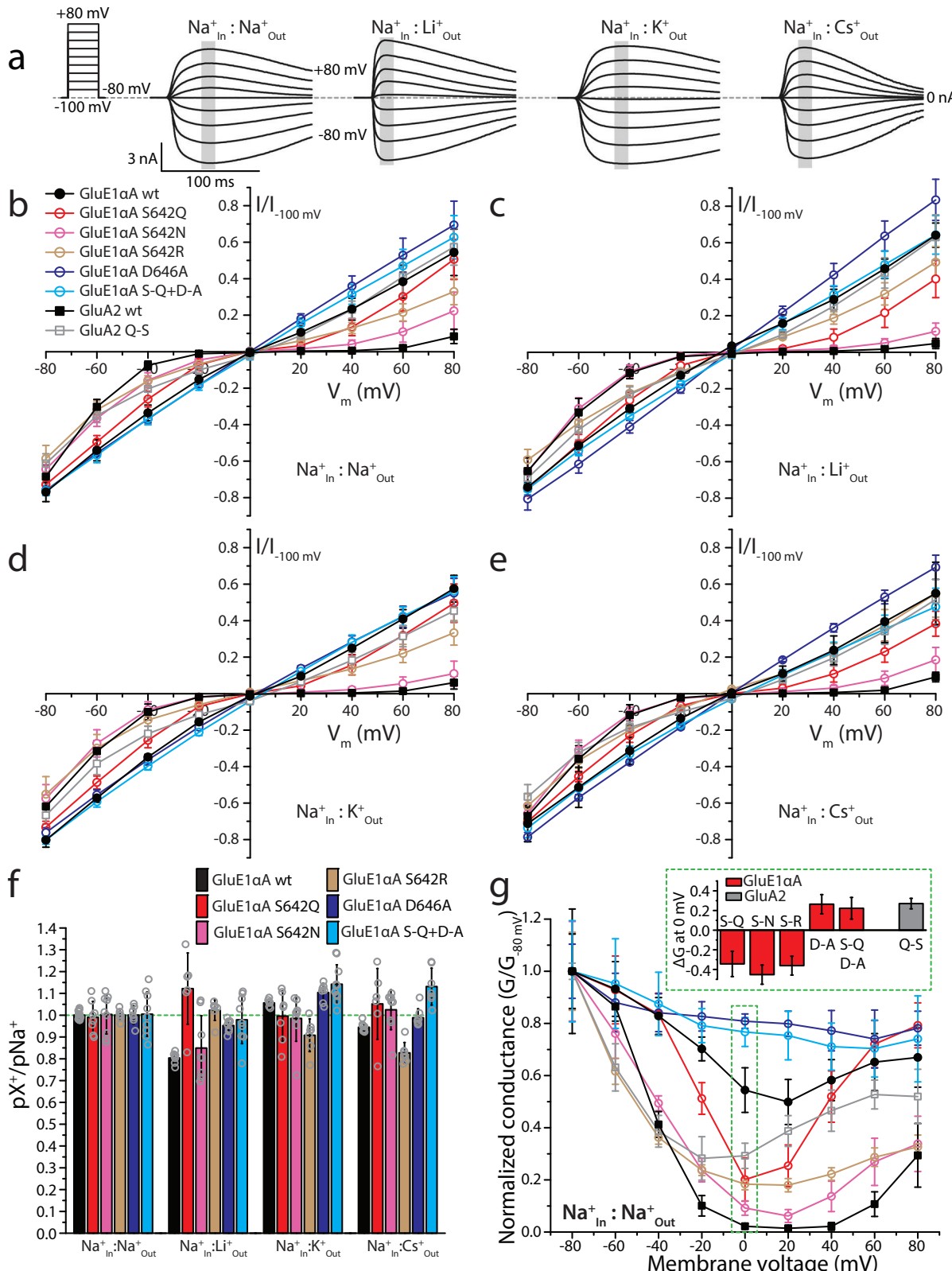

receptor and the serine variant revealed a positive change in normalized conductance of 27.0 ± 5.4%, consistent with reduced polyamine block.

To further elucidate the nature of internal polyamine block in GluE1αA, we performed outside-out membrane patch clamp experiments in wildtype and mutant channels. Excised patches were held at a holding potential of 0 mV, ramped to −100 mV, and then stepped from −100 mV to

100 mV (in 10 mV increments) to observe the degree of block by intracellular spermine (Spm) (30 μM, see Methods). Agonist-evoked membrane currents were elicited by exposing patches to 10 mM glycine for 250 ms (Fig. 6a, d, g, and j). Current-voltage (I-V) relationships were converted to conductance-voltage (G-V) plots and fit with a single permeant blocker model to estimate both the dissociation constant for Spm at 0 mV ($K_{d(0mV)}$)

**Fig. 5 | A non-canonical serine in the Q/R/N site of GluE1αA contributes to non-selective monovalent currents and weak rectification indicative of reduced polyamine block. a** Sample wildtype GluE1αA whole-cell currents elicited by 3 mM glycine recorded at membrane voltages ranging from −80 mV to +80 mV under bi-ionic conditions of 150 mM intracellular Na$^+$ and 150 mM extracellular X$^+$ ions where X is Na, Li, K or Cs. **b** Plot of average normalized current vs. membrane voltages (I-V) under equimolar intracellular and extracellular Na$^+$ conducted by wildtype and pore mutant variants of GluE1αA, the wildtype human GluA2 receptor, and a mutant GluA2 variant bearing a glutamine (Q) to serine (S) mutation in the pore Q/R/N site ($n = 6$–13). **c** Similar I-V plot as **b** but with equimolar internal Na$^+$ and external Li$^+$ ($n = 6$–9). **d** I-V graph with equimolar internal Na$^+$ and

external K$^+$ ($n = 7$–10). **e** I-V plot with equimolar internal Na$^+$ and external Cs$^+$ ($n = 6$–9). **f** Average permeability ratios (pX$^+$/pNa$^+$) ± standard deviation calculated from the I-V data presented in **b** to **e**. **g** Average normalized conductance vs. membrane voltage (G-V) plot derived from the I-V data in **b** for the different receptors and variants under symmetric bi-ionic Na$^+$ conditions. **Inset:** plot of differences in the normalized conductance data for all receptor variants relative to their wildtype counter parts at 0 mV (i.e., values demarked by the green dashed box in the main plot). Differences between average wildtype and corresponding variant conductance data at 0 mV were all significant in paired T-tests ($p \leq 5.5E{-}6$). The legend embedded in **b** pertains to **b** to **e** and **g**.

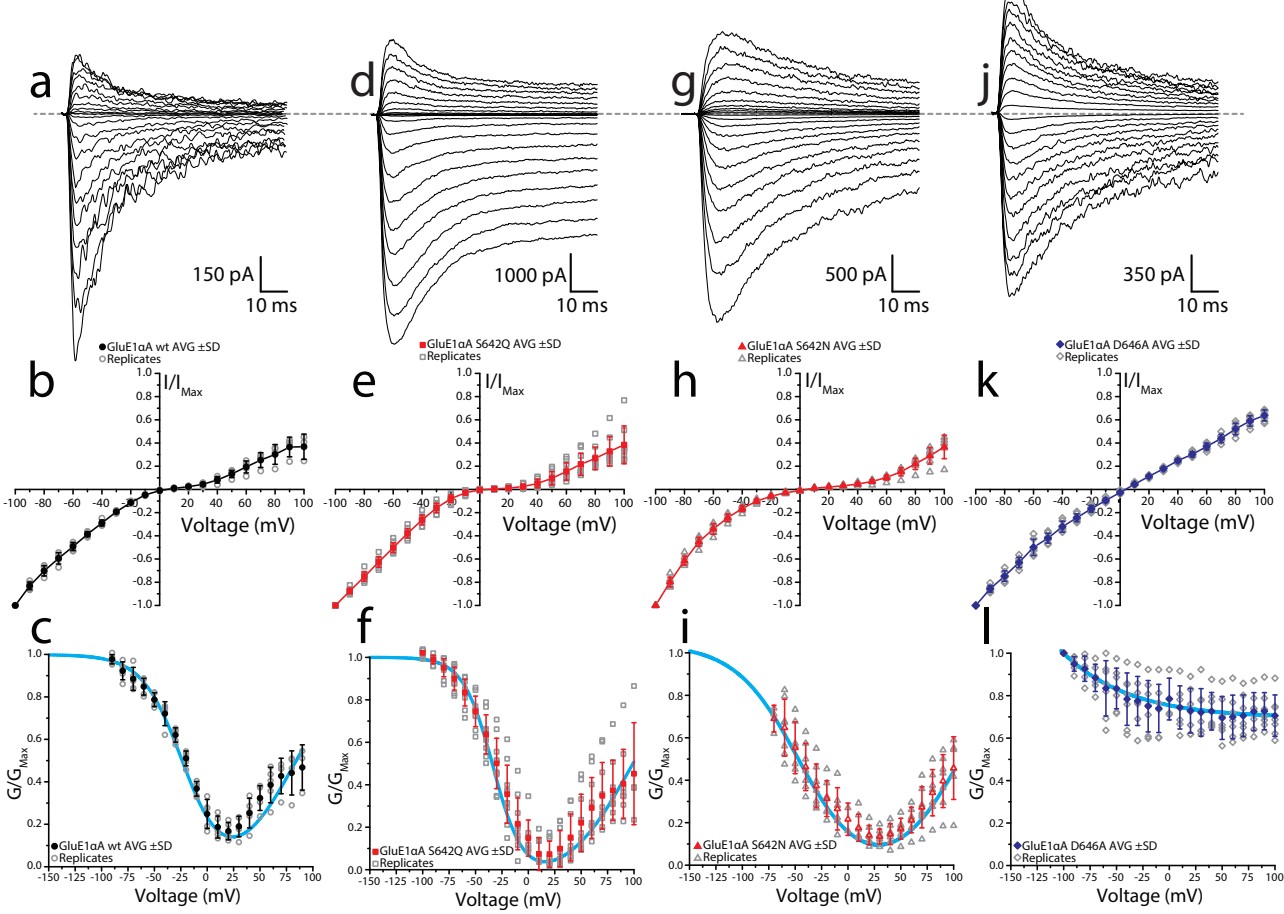

**Fig. 6 | Alterations of Q/R/N and + 4 sites differentially regulate internal poly-amine block in *T. adhaerens* GluE1αA receptors. a, d, g, j** Raw current traces evoked by 250 ms exposure to 10 mM glycine with co-concurrent voltage steps (range, −100 to 100 mV; Δ10 mV). Sample traces correspond to the following: GluE1αA (**a**), GluE1αA S$_{642}$Q (**d**), GluE1αA S$_{642}$N (**g**), and GluE1αA D$_{646}$A (**j**). **b, e, h, k** Mean I-V plots for each receptor complex ( ± standard deviation), normalized to

maximum current response at −100 mV. Data points for single replicates are included in the plot (GluE1αA, $n = 5$; GluE1αA S$_{642}$Q, $n = 10$; GluE1αA S$_{642}$N, $n = 8$; GluE1αA D$_{646}$A, $n = 7$). **c, f, i, l** Mean G-V plots for each receptor complex ( ± standard deviation), normalized to their respective maximal conductance. Cyan lines represent fitting using the single permeant blocker model, apart from GluE1αA D$_{646}$A, which was fit with an exponential equation (described in Methods).

and the voltage dependency of block, as previously described[46,47]. Like results in whole-cell patch clamp experiments, GluE1αA wildtype receptors possessing a noncanonical serine at the Q/R/N site exhibit less polyamine block than that seen in mutant receptors bearing a Q/R/N glutamine, with $K_{d(0mV)}$ values of 11.4 ± 1.9 μM vs. 5.7 ± 3.7 μM, respectively (Fig. 6a, b, and c; Table 1). Comparatively, polyamine block in S$_{642}$N GluE1αA channels exhibited less voltage dependence and slightly lower $K_{d(0mV)}$ than the S$_{642}$Q variant (8.0 ± 2.9 μM) (Table 1). Due to observing large conductance states at negative membrane potentials in this construct, G-V relationships were restricted to -80 mV to appropriately estimate $K_{d(0mV)}$ (Fig. 6g). In agreement with the whole-cell patch clamp experiments, the presence of an alanine at the +4 site of GluE1αA receptors abolishes voltage-dependent

polyamine block producing a linear I-V (Fig. 6k). As such, intrinsic G-V relationships were fit with an exponential function (see Material and Methods) to determine $G_0$, $G_{min}$, and $V_c$ values (14.1 ± 3.3, 12.3 ± 2.6, and −63.1 ± 11.4 mV, respectively) (Fig. 6l). GluE1αA S$_{642}$R experiments were unable to be replicated in outside-out patches due to poor expression, likely due to the presence of the positively charged arginine residue at a key structural region across all four subunits.

Parallel experiments were carried out in human GluA2 homotetramers bearing equivalent Q/R/N and +4 site mutations to compare the regulation of voltage-dependent polyamine block within the broader context of mammalian receptors. Experimental design remained the same, except for using 10 mM glutamate as the agonist used to evoke membrane currents

**Table 1 | Equilibrium spermine block using the single permeant blocker model**

| Receptor | $K_{d(0mV)}$ (µM) | $h$ (mV) | $k$ (mV) | $n$ |
|---|---|---|---|---|
| GluA2(Q) wt | 5.4 ± 2.6 | −18.5 ± 0.7 | 14.3 ± 0.6 | 6 |
| GluA2 $Q_{607}S$ | 36.2 ± 6.7 (***, ª) | −14.0 ± 2.3 | 164.4 ± 31.1 | 4 |
| GluA2(Q) wt + γ2 | 38.4 ± 11.9 | −16.7 ± 1.9 | 18.2 ± 3.0 | 5 |
| GluA2 $D_{611}A$ + γ2 | 168.7 ± 54.6(**) | −58.8 ± 8.8 | 20.8 ± 3.7 | 4 |
| GluE1αA wt | 11.4 ± 1.9 | −18.5 ± 0.6 | 36.5 ± 4.7 | 5 |
| GluE1αA $S_{642}Q$ | 5.7 ± 3.7 (**) | −15.3 ± 0.7 | 28.3 ± 2.1 | 10 |
| GluE1αA $S_{642}N$ | 8.0 ± 2.9 | −26.4 ± 1.3 | 34.9 ± 3.7 | 8 |
| GluE1αA $D_{646}A$ | - | - | - | 7 |

$K_{d(0mV)}$ refers to the dissociation constant of Spm and are listed in micromolar (µM). Voltage dependency of block onset ($h$) and block relief ($k$) are listed as millivolts (mV). The number of patch recordings for each receptor ($n$) is indicated. Comparisons are drawn between mutated constructs and the corresponding wildtype receptor, unless otherwise indicated. GluA2 $Q_{607}S$ $K_{d(0mV)}$: $t_{(8)} = -10.3$, ***$p < 0.001$, unpaired two-tailed Student's t test. $K_{d(0mV)}$ of GluA2 $Q_{607}S$ versus GluE1αA wt: $t_{(7)} = 7.93$, ª$p < 0.001$, unpaired two-tailed Student's t test. GluA2 $D_{611}A$ $K_{d(0mV)}$: $t_{(7)} = -5.23$, **$p = 0.00117$, unpaired two-tailed Student's t test. $K_{d(0mV)}$ of GluE1αA constructs: $F_{(2,19)} = 5.23$, $p = 0.0129$, one-way ANOVA. Tukey's HSD test, **$p = 0.00972$. All values represent mean ± standard deviation.

(Fig. 7a, d, g, and j). GluA2 receptors possessing a serine at the Q/R/N site exhibited appreciably a higher $K_{d(0mV)}$ value compared to the wildtype receptor bearing a glutamine, 36.2 ± 6.7 µM vs. 5.4 ± 2.6 µM respectively, as well as attenuated voltage dependency of Spm relief of block (compare Fig. 7d–f to a–c). Interestingly, the $K_{d(0mV)}$ of GluA2 $Q_{607}S$ receptors is significantly higher than that of wildtype GluE1αA receptors, despite sharing a common Q/R/N site residue (Table 1). Given that the estimated $K_{d(0mV)}$ value of GluE1αA $S_{642}Q$ is not statistically different from wildtype GluA2(Q), the data suggests that the Q/R/N site of GluE1αA can be altered to yield polyamine block properties consistent with GluA2, but not vice-versa. Due to difficulty in obtaining measurable currents with GluA2 $D_{611}A$ channels alone, they were co-expressed with TARP-γ2 to increase receptor trafficking[48]. Their properties were accordingly compared to wildtype GluA2(Q) co-expressed with TARP-γ2. Association of auxiliary proteins was confirmed by noting a larger steady state response and a slowing in the onset of desensitization. G-V analysis reveals that TARP-γ2 attenuates GluA2(Q) polyamine block (Fig. 7g, h, and i), consistent with previous findings[47,49]. In agreement with prior work reported on GluK2 kainate receptors[50], alteration of the +4 site to an uncharged alanine residue eliminated polyamine block despite the presence of glutamine at the Q/R/N site (Fig. 7j, k, and l) and greatly diminished estimates of $K_{d(0mV)}$ of 168.7 ± 54.6 µM (Table 1). In comparison to the equivalent mutation in GluE1αA, GluA2 $D_{611}A$ co-expressed with γ2 exhibits shallow current rectification as opposed to a voltage-independent drop in conductance (Figs. 6l, 7l).

Taken together, the experiments in both *T. adhaerens* GluE1αA receptors and human GluA2 establishes that the Q/R/N and +4 sites of the pore are conserved determinants of intracellular polyamine block in epsilon and AKDF receptors. Furthermore, the emergence of a serine in the Q/R/N site caused diminished, but not abrogated, sensitivity to polyamine regulation in the *T. adhaerens* channel.

## The Q/R/N serine also diminishes Ca²⁺ permeation

Since the Q/R/N site is also known to play a key role in $Ca^{2+}$ permeation, we conducted bi-ionic electrophysiology experiments using an external solution of 4 mM $Ca^{2+}$ and different internal solutions bearing either $Na^+$, $Li^+$, $K^+$, or $Cs^+$ at 100 mM. Eliciting currents with 3 mM glycine through voltage steps between −100 and +80 mV produced large outward currents for the wildtype GluE1αA receptor, compared to comparatively smaller inward

$Ca^{2+}$ currents under all bi-ionic conditions (Fig. 8a). Currents for the wildtype receptor exhibited negative reversal potential values, ranging between −59.2 ± 4.5 mV for $K^+_{In}$:$Ca^{2+}_{Out}$ and −46.9 ± 2.0 mV $Cs^+_{In}$:$Ca^{2+}_{Out}$ (Fig. 8b–e). Converting $E_{Rev}$ values to pCa²⁺/pX⁺ values (where X = Na, Li, K, or Cs) revealed only slight changes in permeability ratios between the four conditions ranging between 0.67 ± 0.11 (for $K^+_{In}$:$Ca^{2+}_{Out}$) to 1.14 ± 0.10 (for $Cs^+_{In}$:$Ca^{2+}_{Out}$) (Fig. 8g and Supplementary Fig. 8a; statistical comparisons provided in Supplementary Table 4b). We made similar recordings of the wildtype human AMPA receptor, finding considerably right-shifted I-V curves, with strong rectification of currents near the reversal potential preventing us from obtaining accurate $E_{Rev}$ values (Fig. 8b–e and Supplementary Fig. 8a). Notably different between the human and *T. adhaerens* iGluRs is the ratio of inward $Ca^{2+}$ current at −100 mV vs. outward monovalent current at +80 mV. Specifically, while the inward $Ca^{2+}$ current ranged between 26.3 ± 5.6% and 50.6 ± 8.8% of the outward monovalent current for GluE1αA (for the conditions $K^+_{In}$:$Ca^{2+}_{Out}$ and $Cs^+_{In}$:$Ca^{2+}_{Out}$, respectively), the relative inward $Ca^{2+}$ current for GluA2 ranged between 175.1 ± 25.3% for $Na^+_{In}$:$Ca^{2+}_{Out}$, and 892.1 ± 107.0% for $Li^+_{In}$:$Ca^{2+}_{Out}$; the latter attributable to very small outward currents when $Li^+$ was present as the outward-permeating ion (Fig. 8b–e and Supplementary Fig. 8a). In G-V plots normalized to −80 mV, a difference in inward $Ca^{2+}$ permeation is also apparent between GluE1αA and GluA2. That is, while the normalized inward $Ca^{2+}$ conductances of GluA2 at −80 mV range between 77.4 ± 5.2% ($Na^+_{In}$:$Ca^{2+}_{Out}$) and 382.4 ± 9.7% ($Li^+_{In}$:$Ca^{2+}_{Out}$), those for GluE1αA were much lower ranging between 27.9 ± 4.5% ($K^+_{In}$:$Ca^{2+}_{Out}$) and 49.6 ± 5.7% ($Cs^+_{In}$:$Ca^{2+}_{Out}$; Fig. 8g and Supplementary Fig. 8b–e). Altogether, these observations indicate that the *T. adhaerens* GluE1αA receptor bearing a serine in the Q/R/N site is less permeable to $Ca^{2+}$ compared to GluA2.

Next, we tested whether a serine residue in the Q/R/N site of GluA2 imparts similar low $Ca^{2+}$ permeation. Despite our inability to calculate accurate reversal potentials for this variant due to its strong rectifying currents, we observed clear leftward shifts in the I-V curves under all bi-ionic conditions, along with marked reductions in the ratio of inward $Ca^{2+}$ current at −100 mV relative to outward monovalent current at +80 mV (Fig. 8b–e), concurrent with reductions in inward $Ca^{2+}$ conductance at negative voltages in the G-V plots (Fig. 8g and Supplementary Fig. 8b–e). Thus, a serine residue in the Q/R/N site significantly reduces $Ca^{2+}$ vs. monovalent permeability in GluA2.

We also conducted experiments to determine whether the AMPA/kainate/NMDA-like Q/R/N variants of GluE1αA exhibit enhanced $Ca^{2+}$ permeation properties. In agreement with expectations, both glutamine and asparagine in the Q/R/N site led to strong rightward shifts in the I-V curves under all bi-ionic conditions (Fig. 8b–e), and corresponding strong increases in pCa²⁺/pX⁺ ratios ranging between 6.6 ± 0.6 ($Cs^+_{In}$:$Ca^{2+}_{Out}$) and 10.8 ± 0.5 ($Li^+_{In}$:$Ca^{2+}_{Out}$) for the $S_{642}Q$ variant, and between 3.1 ± 0.2 ($K^+_{In}$:$Ca^{2+}_{Out}$) and 4.1 ± 0.8 ($Cs^+_{In}$:$Ca^{2+}_{Out}$) for the $S_{642}N$ variant (Fig. 8f). There were also general increases in normalized inward $Ca^{2+}$ conductance at negative voltages, especially for $S_{642}Q$ (Fig. 8g and Supplementary Fig. 8b–e), as well as U-shapes G-V curves consistent with increased polyamine block. Altogether, this data supports a significant increase in $Ca^{2+}$ permeability and conductance when the Q/R/N serine is mutated to resemble mammalian iGluRs. In contrast, the $S_{642}R$ mutation, despite causing mild rightward shifts in I-V plot reversal potentials and mild increases in pCa²⁺/pX⁺ values, produced very small inward $Ca^{2+}$ currents at −100 mV relative to outward monovalent currents at +80 mV (Fig. 8b–e), and significantly diminished normalized inward $Ca^{2+}$ conductance at negative voltages (Fig. 8g and Supplementary Fig. 8b–e). These observations are consistent with those made for homotetrameric mammalian AMPA and kainate receptors bearing four arginine residues in the Q/R/N sites, which are poorly conductive to cations[51], perhaps due to repulsion of permeating cations by the positively-charged arginine side chains[52].

Lastly, and consistent with our monovalent vs. monovalent experiments, the $D_{646}A$ variant of GluE1αA produced more linear I-V curves compared to wildtype, attributable to loss of voltage-dependent polyamine block

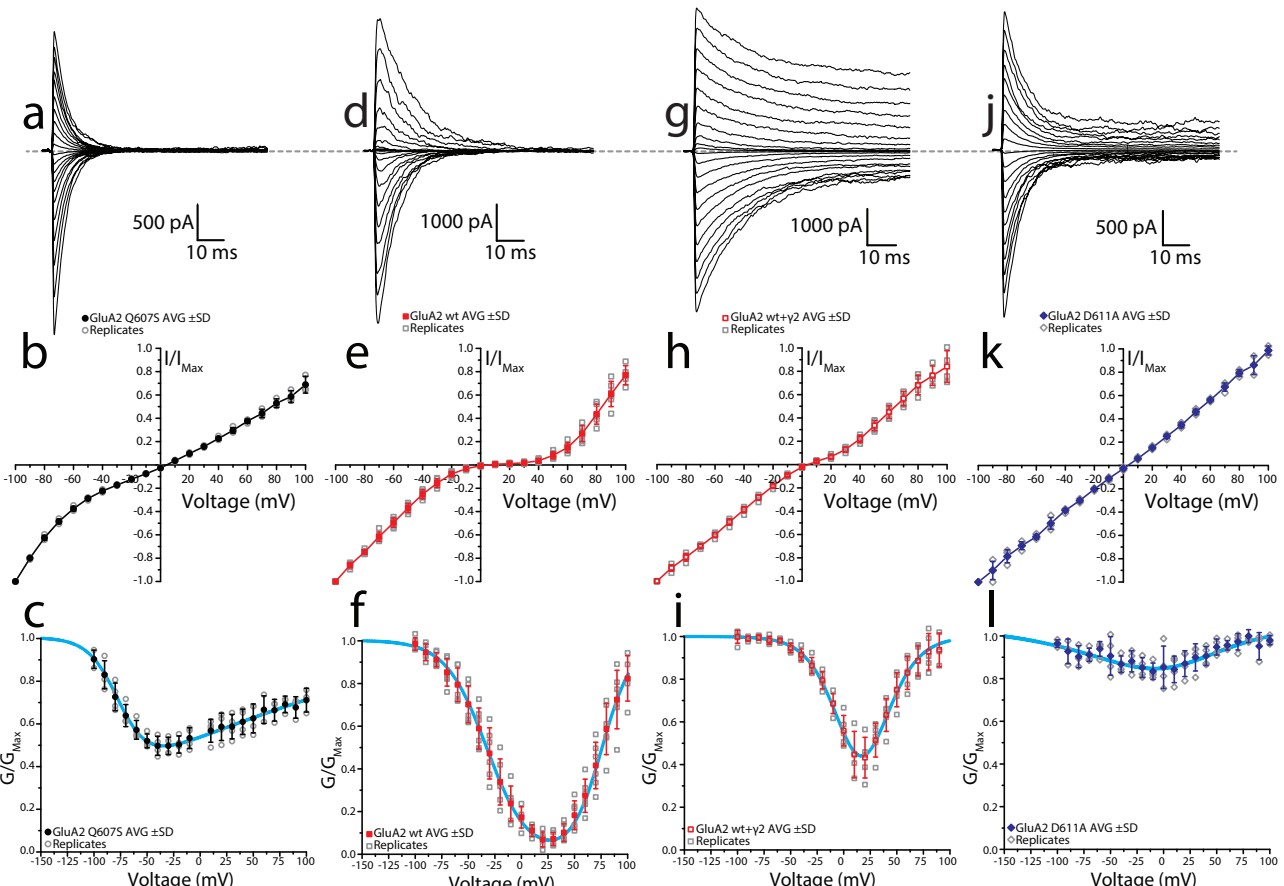

**Fig. 7 | Alterations of Q/R/N and + 4 sites differentially regulate internal polyamine block in human GluA2(Q) receptors. a, d, g, j** Raw current traces evoked by 250 ms exposure to 10 mM glutamate with co-concurrent voltage steps (range, −100 to 100 mV; Δ10 mV). Sample traces correspond to the following: GluA2 $Q_{607}S$ (**a**), GluA2(Q) (**d**), GluA2(Q) + γ2 (**g**), and GluA2 $D_{611}A$ + γ2 (**j**). **b, e, h, k** Mean I-V plots for each receptor complex ( ± standard deviation), normalized to maximum current response at −100 mV. Data points for single replicates are included in the plot (GluA2 $Q_{607}S$, $n = 4$; GluA2(Q), $n = 6$; GluA2(Q) + γ2, $n = 5$; GluA2 $D_{611}A$ + γ2, $n = 4$). **c, f, i, l** Mean G-V plots for each receptor complex, normalized to their respective maximal conductance. Cyan lines represent fitting using the single permeant blocker model.

(Fig. 8b–e), also apparent in the more linear G-V plots for this variant (Fig. 8g and Supplementary Fig. 8b–e). This mutation also produced small but statistically significant changes in calcium vs. monovalent permeation, with lower $pCa^{2+}/pX^+$ values compared to wildtype for all bi-ionic conditions, respectively, with $0.81 ± 0.14$ vs. $0.61 ± 0.07$ for $Na^+_{In}$:$Ca^{2+}_{Out}$, $1.01 ± 0.28$ vs. $0.59 ± 0.05$ for $Li^+_{In}$:$Ca^{2+}_{Out}$, $0.67 ± 0.11$ vs. $0.61 ± 0.04$ for $K^+_{In}$:$Ca^{2+}_{Out}$, and $1.14 ± 0.10$ vs. $0.99 ± 0.06$ for $Cs^+_{In}$:$Ca^{2+}_{Out}$ (Fig. 8f and Supplementary Table 4b). Combined with the $S_{642}Q$ mutation, the $D_{646}A$ mutation had no effects in the presence of $Na^+$ and $K^+$ relative to $S_{642}Q$ on its own, but did reduce $Ca^{2+}$ permeability in the presence of the non-physiological ions $Li^+$ (i.e., $10.79 ± 0.54$ for $S_{642}Q$ vs. $7.08 ± 0.54$ for $S_{642}Q$ plus $D_{646}A$) and $Cs^+$ ($7.85 ± 0.56$ for $S_{642}Q$ vs. $6.46 ± 1.14$ for $S_{642}Q$ plus $D_{646}A$) (Fig. 8f and Supplementary Table 4b). Altogether, these experiments corroborate our previous observations that the presence of a serine in the Q/R/N site diminishes polyamine block, and in addition, reduces $Ca^{2+}$ permeation.

## Insights into polyamine regulation of GluE1αA by modelling and extracellular perfusion of polyamines

We used homology modelling and docking to explore the energy properties of spermine in the pore of the wildtype receptor bearing a serine residue in the Q/R/N site, and the asparagine $S_{642}N$ variant which exhibited significantly enhanced polyamine block (Figs. 5, 6, and 8). To start, we used AlphaFold 2 to predict the tetrameric structure of both variants lacking their N-terminal domains (i.e., $V_{416}$ to $I_{921}$ of NCBI accession number PP886186). Both predicted structures resembled a closed pore

conformation, perhaps expected given that most experimental structures used for training the AlphaFold artificial neuronal network were in the closed conformation. Because we were interested in exploring how spermine block occurs in the open, conducting state, we modelled the open-pore domain for our spermine docking experiments. To build such models, we isolated the pore domain of the predicted GluE1αA structures ($A_{554}$ to $V_{682}$ and $K_{833}$ to $I_{921}$ of PP886186) and 3D-aligned these to the cryo-EM structure of the GluA2 receptor in the open state (PDB number: 6o9g[45];). This was achieved by minimizing root mean square (RMS) deviations of alpha carbon atoms in sequentially matching positions of the first pore-loop helix of the M2 domains of the two receptors (as reported previously[53]). To reduce 3D misalignment of key P-loop residues in the Q/R/N and +4 sites, we then Monte Carlo-minimized the RMS deviations between matching alpha carbons in the M1 and M3 of the AlphaFold models and the GluA2 cryo-EM structure. Finally, we used the in silico opened models of the wildtype and $D_{646}A$ variants of GluE1αA to predict energetically optimal binding modes of spermine in the pore (Fig. 9a–d). This was done by using the ZMM GRID module (see Materials and Methods) to pull spermine from the cytoplasm to the extracellular side of each GluE1αA variant structure with steps of 0.5 angstroms, and Monte Carlo energy minimizing the structure at each step. Translation of the spermine central atom along the pore axis (its z-coordinate) was frozen, backbone alpha carbon atoms were restrained by "pins", and all other generalized coordinates of the system were free to move during energy minimizations (Supplementary Fig. 9a, b). Energy plots of spermine-receptor interactions reveal declining electrostatic

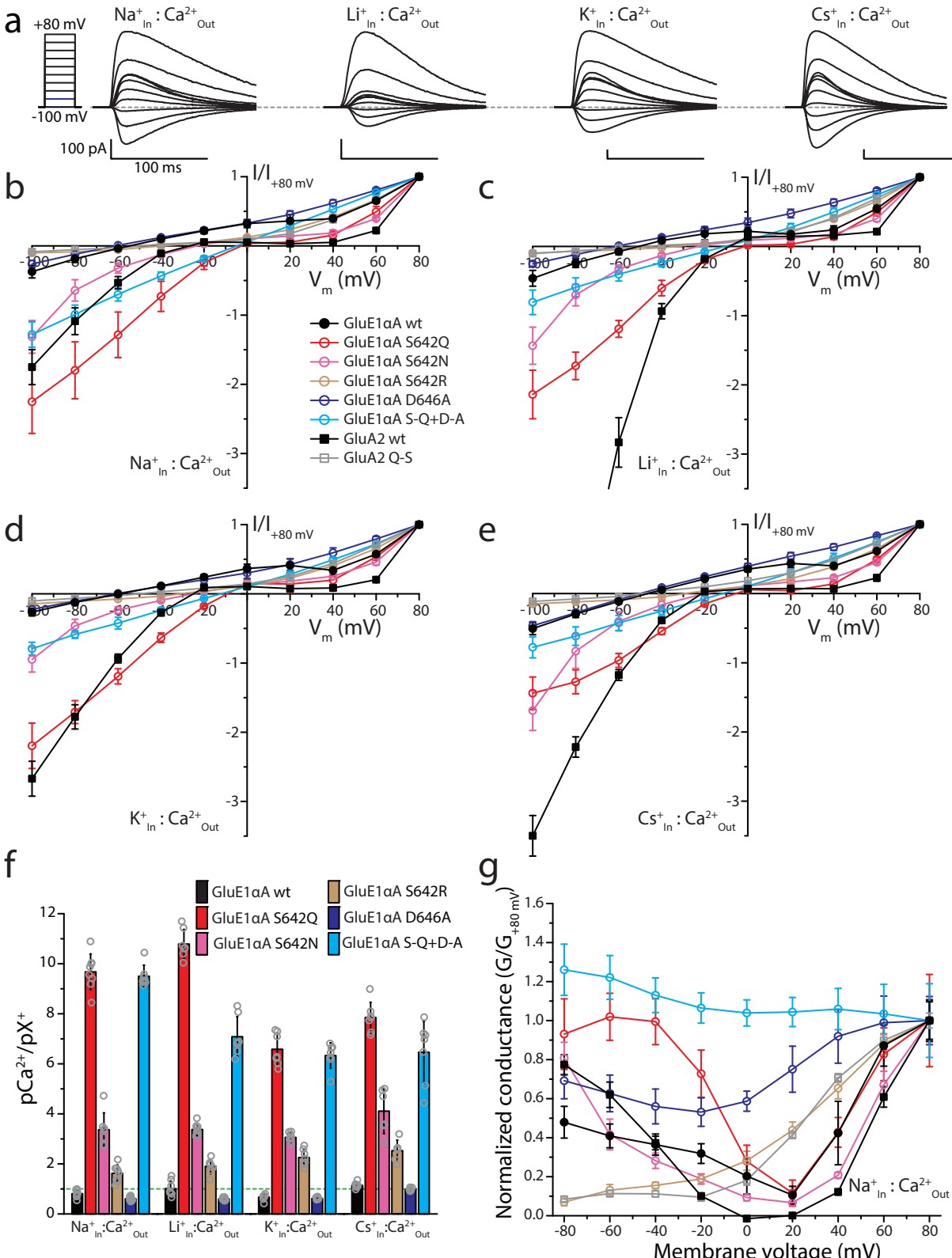

**Fig. 8 | The Q/R/N serine renders reduced Ca²⁺ permeation for GluE1αA compared to variants with canonical Q, N, and R residues. a** Sample wildtype GlueE1α whole-cell currents recorded at membrane voltages ranging from −100 to +80 mV under bi-ionic conditions of 4 mM extracellular Ca²⁺ and 100 mM intracellular X⁺ where X denotes Na, Li, K or Cs. **b** Plot of average normalized current vs. membrane voltages (I-V) under 4 mM extracellular Ca²⁺ and 100 mM intracellular Na⁺ conducted by wildtype and pore mutant variants of GluE1αA, the wildtype human GluA2 receptor, and a mutant GluA2 variant bearing a glutamine to serine mutation in the QRN site (GluA2 Q607S; n = 7-11). **c** Similar I-V plot as in **b** but with 100 mM internal Li⁺ instead of Na⁺ (*n* = 7–8). **d** I-V plot with 100 mM internal K⁺ (*n* = 7–8). **e** I-V plot with 100 mM internal Cs⁺ (*n* = 6–8). **f** Average permeability ratios (pCa²⁺/pX⁺) ± standard deviation calculated from the I-V data presented in **b** to **e**. **g** Average normalized conductance vs. membrane voltage (G-V) plot derived from the I-V data in **b** for the different receptors and variants under external Ca²⁺ and internal Na⁺ conditions. The legend embedded in **b** pertains to **b** to **e** and **g**.

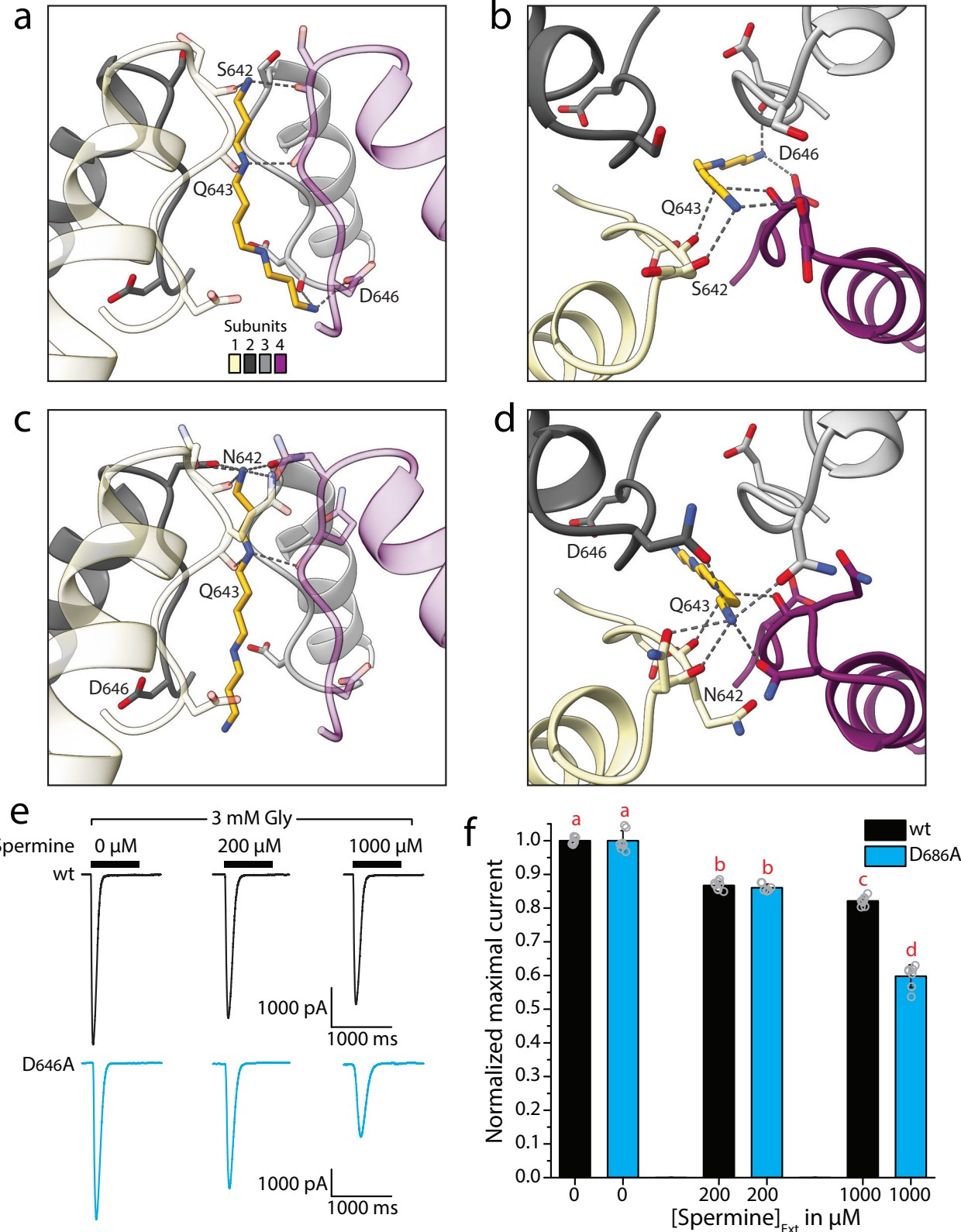

**Fig. 9 | GluE1αA pore residues contribute to intracellular and extracellular regulation by polyamines. a** Homology model and Monte Carlo-minimized structure of spermine in the open GluE1αA pore domain (side view, step 32). **b** Top-down view of the structure in **a**. **c** Homology model and Monte Carlo-minimized structure of spermine in the open pore domain of the $S_{642}N$ variant of GluE1αA (step 34). **d** Top-down view of the structure in **c**. In panels **a** to **d**, predicted hydrogen bonds between pore-loop residues of the subunits and spermine are depicted by dashed lines. **e** Sample wildtype and $D_{646}A$ currents elicited with glycine co-applied with increasing concentrations of perfused (extracellular) spermine. **f** Bar plot of normalized currents elicited by 3 mM glycine with and without co-applied spermine. Letters above the bars denote statistically significant differences among and between conditions determined by post hoc Tukey tests ($p < 0.005$) after One Way ANOVAs (F = 385.66, $p < 1E-14$).

(ELRT) and non-bonded (ELRN) energy components for both the wildtype and $S_{642}N$ variants as spermine enters the pore, while the desolvation energy (ELRD) increased reflecting increasing dehydration. Despite the latter, both receptors had decreasing total energy values (ELRT) as spermine moved into the pore, with most favourable positions at step 32 for the wildtype receptor and step 34 for the $S_{642}N$ variant (Supplementary Fig. 9b and c). In the wildtype receptor, the cytoplasmic (lower) amino group of spermine forms strong hydrogen bonds with the +4 $D_{646}$ residues of subunits 3 and 4 (Fig. 9a, b), which contribute the most favorable energies of the receptor-ligand interactions, while the same aspartates from subunits 1 and 2 also contribute (Table Supplementary 3d, e). Similarly, all four $D_{646}$ residues in the $S_{642}N$ variant contribute favorable energies for interactions with spermine, though with smaller values than wildtype (Fig. 9c, d; Supplementary Table 3d–e). These computations are overall consistent with our experimental data where $D_{646}A$ variants of GluE1αA exhibited a near complete loss of polyamine block evident as negligible current rectification (Figs. 5, 6 and 8). Both the wildtype and mutant structures are predicted to have electrostatic interactions between upper/central amino group in the spermine molecule and highly conserved glutamine residues one position downstream of the Q/R/N site (Fig. 1b), in both cases involving subunits 1 and 4 (Fig. 9a–d). Comparing total energy scores, we note a more negative score for spermine in step/position 34 of the $S_{642}N$ variant, compared to the wildtype receptor at step 32, the former having a total receptor-ligand score of −20.3 kcal/mol, and the latter of −16.4 kcal/mol (Supplementary Table 3d). In part, this marked energy difference is attributable to the Q/R/N asparagine residues which collectively contribute −6.3 kcal/mol to the total spermine-receptor energy, forming numerous predicted hydrogen bonds with the extracellular/upper ammonium group of spermine, while the Q/R/N serine residue in subunit 1 of the wildtype receptor contributes only −0.8 kcal/mol, and the same serine in subunit 3 provides a net repulsive energy of 0.4 kcal/mol (Supplementary Table 3d).

As a final set of experiments, we used perfusion to determine whether spermine applied extracellularly causes changes to ligand-induced macroscopic currents of GluE1αA, as has been reported for select mammalian iGluRs (reviewed in ref. 3). Application of 200 µM spermine caused a modest reduction in peak inward current of 13.3 ± 1.3% relative to currents elicited by 3 mM glycine alone, while 1000 µM spermine reduced peak current by 17.9 ± 1.3% (Fig. 9e, f). Since this observed block could be due to spermine molecules entering the pore from the extracellular side, we also tested the $D_{646}A$ variant, reasoning that it should have altered sensitivity if the observed effects are due to extracellular polyamines entering the pore. Unexpectedly, 1000 µM spermine caused a more pronounced block for the $D_{464}A$ mutant, reducing peak current by 40.2 ± 3.4%, while the percent block caused by 200 µM spermine was indistinguishable from that of the wildtype receptor (i.e., 14.0 ± 1.1%; Fig. 9e, f). Although the reason for this is unclear, it is notable that neutralization of the +4 aspartate residues via alanine substitution removes a ring of four negatively charged carboxyl side groups which could facilitate the transfer of polyamines across the pore, whereby in their absence, extracellular polyamines moving through the channel could become trapped in upper regions of the pore, hence blocking inward current.

## Discussion

### An updated phylogeny of placozoan iGluRs

Our phylogenetic analysis is consistent with several recent studies and provides several new insights, particularly into the evolution of iGluRs within the phylum Placozoa. One notable insight is the one-to-one orthology of AKDF receptors among the four examined species, despite their deep branch separation (Fig. 1b). Their cell-type mRNA expression is also similar between species, with most receptors having a top cellular expression in digestive lipophil cells. Epsilon receptors on the other hand appear to have undergone lineage-specific duplications and losses, resulting in much more phylogenetic diversity, combined with more heterogeneity in cellular expression across species (Fig. 1b). Consistent with the species phylogeny[25], *T. adhaerens* and *Trichoplax* species H2 share more

orthologous receptors with each other, as do *H. hongkongensis* and *C. collaboinventa*, and these are often expressed in the same cell type. Of note, during our manual curation and analysis of placozoan iGluRs we identified a small number of epsilon receptor sequences from each species that were too fragmented to include in our phylogenetic analysis. Indeed, as more and improved gene data becomes available, for these or perhaps other placozoan species, more complete iGluR sequences might emerge that will help further resolve the phylogeny of placozoan epsilon receptors. Nonetheless, the three strongly supported clades of epsilon receptors delineated in our phylogenetic analysis, bearing deep and strongly supported roots, will likely withstand the addition of new sequences, which in turn, can be named according to our proposed nomenclature.

Interestingly, like AKDF receptors, the GluE3α and GluE3β receptors show enriched expression in lipophil cells, while GluE1 receptors tend to be enriched in neuron-like peptidergic cells (Fig. 1b). Currently, our understanding about the physiological functions of iGluRs in non-bilaterian animals is very limited. In cnidarians, glutamate is not considered to be a major neurotransmitter in synapses, although it elicits various behavioral responses in select species, mainly associated with feeding including chemotaxis, discharge of cnidocytes used for hunting and defense, and muscle contractions (reviewed in ref. 1). Indeed, the enriched expression of iGluRs in placozoan lipophil cells, which are involved in feeding, might indicate a similar function in placozoans, and perhaps more broadly. That is, that a primordial function of iGluRs, before the emergence of synapses, was for sensing amino acids in the environment to identify food sources and mediate feeding behaviors. Instead, the expression of iGluRs in neuron-like peptidergic cells might reflect roles more akin to neural signaling, where for example the receptors could serve to depolarize cells in response to ligands present in the environment, triggering secretion of peptides that regulate behavior. As noted earlier, placozoans elicit numerous behavioral responses to applied transmitters including glycine, glutamate, and GABA[15,16]. However, like cnidarians, there is no evidence suggesting that placozoans actively secrete these substances into the environment in a regulated manner to affect other cells or coordinate behavioral responses, although some cells are reported to possess pale vesicles, in addition to dense core, the former consistent with small-molecule transmitters[54,55]. Although possible, something that needs to be reconciled is how effective extracellular concentrations are reached, given that placozoans lack synapses[55], or tight/septate junctions between epithelial cells that would prevent secreted molecules from quickly diffusing out of the animal interior[56,57].

Prior to this study, the only placozoan iGluR to be functionally characterized in vitro is the AKDF1 receptor from *T. adhaerens*[31]. As noted, this receptor lacks an aspartate residue in its ligand binding domain that in other receptors interacts with the backbone amino group of amino acid ligands ($D_{705}$), leading to constitutive leak currents that are blocked by various ligands, most prominently glycine as predicted by its ligand-binding sequences. This same aspartate to tyrosine adaptation is evident in all placozoan AKDF1 orthologues (Fig. 1b), indicating this is an ancestral adaptation within the phylum. Interestingly, several Epsilon receptors also lack ligand backbone interacting residues, including members of the GluE1 clade which also lack $D_{705}$ residues. These same receptors also have sequence differences in their ligand binding domains that prevent prediction of their ligand specificity. As such, this clade of receptors might have unique ligand binding properties. Lastly, it is evident that evolutionary changes in ligand specificity are apparent within both the AKDF and epsilon receptors in placozoans, consistent with predictions of ligand specificity changes among all major lineages of metazoan iGluRs[11].

### Insights into iGluR ligand-specificity

Ligand specificity of iGluRs has been subject to considerable study, spurred by early realizations that different types of receptor subunits respond differently to natural ligands (reviewed in ref. 3). Our electrophysiological characterization of the wildtype GluE1αA revealed a broad ligand specificity, with sub-millimolar sensitivity to glycine, alanine, and D-/L-serine, and millimolar sensitivity to valine (Fig. 2). In its preference for glycine over

glutamate, GluE1αA resembles other homomeric epsilon receptors from *B. lanceolatum* (GluE1) and *M. leidyi* (ML032222a) but differs from a second *M. leidyi* receptor (ML05909) that is activated by both glutamate and glycine[5,8]. Notable is that alanine was the most potent activator of GluE1αA, a hydrophobic amino acid that also activates the glycine-sensitive NMDA1/GluN1 subunit[58], as well as the divergent GluR1 receptor from the rotifer *Adineta vaga*, a highly atypical receptor that more resembles bacterial iGluRs phylogenetically, structurally, and in its selectivity for K$^+$ over Na$^+$ ions[59,60]. This receptor, which is also activated by glutamate but not glycine, is more broadly sensitive to hydrophobic amino acids compared to GluE1αA and GluN1, exhibiting strong activation by methionine, cysteine, and moderate activation by phenylalanine[60]. Interestingly, X-ray structures and functional studies uncovered unique ligand properties of the *A. vaga* GluR1 receptor, whereby chloride ions act as co-activators for hydrophobic amino acids by forming surrogate interactions with residues in the ligand binding domain that normally associate with the glutamate side chain[60]. Since the wildtype *T. adhaerens* receptor did not respond to glutamate, while the m3 variant responded to glutamate but not alanine (Fig. 3b), it seems unlikely that a similar mechanism accounts for the activation of GluE1αA by alanine and the also hydrophobic amino acid valine (Fig. 2b, c).

As noted in the results, Ramos-Vicente et al.[11] reviewed known structural determinants for ligand-binding by various iGluRs to propose that five key amino acids within the ligand binding domain are predictive for glutamate vs. glycine/D-serine specificity (Fig. 3a). Based on their analysis of these five positions, they predicted that both glutamate and glycine receptors exist within each major lineage of metazoan iGluRs: AKDF, NMDA, epsilon, and lambda[11]. Their prediction certainly proved correct for the placozoan GluE1αA receptor characterized in this study (Fig. 2a), as well as other non-vertebrate iGluRs that have been functionally characterized such as the glycine-sensitive AKDF1 receptor also from *T. adhaerens*[31], the glycine-activated GluE1 epsilon receptor from *Branchiostoma lanceolatum*[5], and the glutamate-activated kainate 1D receptor from *Drosophila melanogaster*[61]. To the best of our knowledge, ours is the first study to test this prediction model by seeking to switch ligand specificity of an iGluR.

By mutating just three amino acids in the ligand binding domain, we were able to completely switch the ligand specificity of GluE1αA from glycine/serine to glutamate. In fact, the m2 variant bearing just two mutations (S$_{653}$G and I$_{655}$T) changed the specificity mostly towards glutamate, while the third mutation, F$_{732}$Y, increased sensitivity to glutamate (Fig. 3b and c), almost completely abolished glycine sensitivity (Fig. 3b, c) and caused nascent sensitivity to AMPA (Fig. 4a, b). In our models, this mutated tyrosine residue forms salt bridge with reside D$_{705}$, which in the wildtype channel forms key contacts with glycine, leading to reduced binding energy between D$_{705}$ and glycine in the mutant receptor (Supplementary Table 3). Perhaps, this contributes to the near complete loss of glycine sensitivity of the m3 triple mutant. In addition, the F$_{732}$Y mutation introduces a hydroxyl group into the ligand binding pocket, increasing its polarity and perhaps helping to attract the side chain carboxyl group of glutamate into the pocket. This latter mutation would also diminish the hydrophobicity of the pocket, perhaps decreasing binding affinity for hydrophobic amino acid ligands, consistent with the complete loss of valine sensitivity by the F$_{732}$Y variant of GluE1αA tested at 10 mM (Fig. 3b and c). However, this same variant also became slightly more sensitive to alanine, relative to the wildtype receptor, indicating more nuanced factors contribute to receptor activation by hydrophobic amino acids.

According to the ligand specificity hypothesis, when position 655 in the ligand binding domain is hydrophobic (i.e., V/L/A/I/P), there is a preference for glycine, while a polar threonine contributes to glutamate specificity[11]. In wildtype GluE1αA, this equivalent position bears an isoleucine, which we mutated to a threonine in the m2 and m3 variants (Fig. 3a). Like the F$_{732}$Y mutation, the I$_{655}$T mutation would introduce a hydroxyl group into the ligand-binding pocket increasing its polarity, and in our models, this residue forms hydrogen bonds with the glutamate side chain (Fig. 3i and Supplementary Fig. 6d; Supplementary Table 3). Additionally, mutation of serine at position 653 to glycine would enlarge the binding pocket, perhaps better

accommodating the larger glutamate side chain (Fig. 3h, I and Supplementary Fig. 6c, d).

Interestingly, the three mutations that switched the ligand specificity of GluE1αA from glycine to glutamate also disrupted sensitivity to alanine, L-and D-serine, and valine, making the receptor more similar to AMPA and kainate receptors that are more exclusively gated by glutamate over other amino acids (reviewed in ref. 3). If this observation extrapolates as a general feature of iGluRs, it may point to one of the selective pressures that drove evolutionary changes in ligand specificity, with conceivable advantages for detecting just glutamate in some contexts (i.e., in synapses or the environment), or a broader set of amino acids in other contexts. Also interesting was that the mutations that switched ligand specificity of GluE1αA also imposed nascent sensitivity to AMPA (Fig. 4a, b) and increased sensitivity to the blocker CNQX (Fig. 4c–e), further likening the placozoan receptor to vertebrate AMPA and kainate receptors. Together, these observations suggest that structural features of the ligand binding domain that promote exclusive glutamate sensitivity also increase a receptor's affinity for pharmacological compounds that are selective for such channels. However, there are apparently structural features that do not overlap for binding natural ligands vs. pharmacological compounds, since compounds that discriminate between the different vertebrate receptors do not necessarily do so for their invertebrate counterparts. For example, the heteromeric iGluRs that are found in the fruit fly neuromuscular junction, which are phylogenetically classified as kainate receptors, are insensitive to kainate[62], while the AMPA receptor GLR-1 from the worm *Caenorhabditis elegans* is activated by kainate[63].

Lastly, it is worth noting that epsilon receptors from ctenophores evolved a unique mechanism for glycine specificity compared to other metazoan iGluRs, including epsilon receptors from other species. Indeed, ctenophore epsilon subunits have different residues at positions 653, 655, and 704 than those associated with glycine specificity[11], but still bind glycine[8,9]. Crystal structures of ML032222a and PbiGluR3 from the respective ctenophore species *M. leidyi* and *P. bachei* revealed a salt bridge between the two lobes of the LBD that traps glycine in the binding pocket, rendering a unique mechanism for glycine binding[8,9].

## Insights into ion permeation and polyamine regulation

Our phylogenetic analysis revealed that all placozoan GluE1 type receptors bear atypical Q/R/N residues of serine, leucine, or alanine, while the remaining epsilon receptors, and the four AKDF receptor types, bear 'canonical' glutamine residues also found in AMPA and kainate receptors (Fig. 1b). Similar variability in the Q/R/N site is apparent among the epsilon receptors from the ctenophore species *M. leidyi*, most bearing a glycine[8]. This variability extends to epsilon receptors from cephalochordates, for example GluE1 from *B. lanceolatum* which bears a phenylalanine[5], and indeed to all classes of metazoan iGluRs[5]. Perhaps the least variable among these are the AKDF receptors, that like the *T. adhaerens* AKDF receptors, as well as AMPA, and kainite receptors, tend to have a glutamine in the Q/R/N site[5]. Nonetheless, the general prevalence of glutamine in the Q/R/N site of AKDF receptors, and its frequent occurrence in epsilon receptors, suggests this residue was acquired through common descent. By extension, epsilon receptors might have undergone evolutionary changes that relaxed selective pressures for retaining a Q/R/N glutamine, allowing for diversification in ion conducting properties and regulation by polyamines.

Our characterization of GluE1αA revealed that the presence of a serine in the Q/R/N site, which has a shorter side chain than glutamine and asparagine and bears a hydroxyl rather than an amino group, had minimal consequences on monovalent vs. monovalent selectivity (Fig. 5 and Supplementary Fig. 7). However, compared to glutamine and asparagine, the Q/R/N serine significantly diminished Ca$^{2+}$ permeation, and this was also apparent for the human GluA2 receptor bearing a Q to S mutation in the Q/R/N site (Fig. 8 and Supplementary Fig. 8). Conversely, mutating the GluE1αA Q/R/N serine to glutamine and asparagine increased Ca$^{2+}$ permeation by roughly 10-fold and 3-fold, respectively (Fig. 8f), perhaps

attributable to stronger coordination of $Ca^{2+}$ ions facilitated by the longer side chains bearing carbonyl oxygen atoms.

Physiologically, it is notable that the GluE1 clade of epsilon receptors, all with variable Q/R/N sites, tend to be enriched in peptidergic cells (Fig. 1b). Based on single cell transcriptome analysis these resemble primordial neurons both in their expression of genes involved in neural development, and genes required for regulated secretion of signaling molecules including voltage-gated calcium channels and machinery for presynaptic exocytosis[25]. In these cells, expression of a receptor with a Q/R/N serine would permit ligand-induced depolarization of the membrane while diminishing the added effect of $Ca^{2+}$ signaling, perhaps serving a functional purpose. Such an evolutionary change is not without precedent, occurring for example in voltage-gated sodium ($Na_V$) channels. Here, ancestral $Na_V$ channels that bear aspartate/glutamate residues in the selectivity filter (i.e., pore motifs of DEEA) are highly permeable to $Ca^{2+}$, while bilaterian $Na_V1$ channels evolved a lysine residue in domain III (i.e., DEKA motifs) to become highly selective for $Na^+$ over $Ca^{2+}$. Interestingly, a parallel adaptation occurred in cnidarian $Na_V2$ channels, but in domain II (i.e., DKEA motifs), leading to convergent evolution of sodium selectivity[64].

Our experiments also indicate that a Q/R/N serine, in both GluE1αA and GluA2, diminishes but does not abrogate polyamine regulation. Instead, mutation of this residue in the *T. adhaerens* receptor to glutamine, asparagine, or arginine increased current rectification, most prominent for the $S_{642}N$ variant (Figs. 5, 6, and 8). According to our modeling, this higher degree of rectification is consistent with higher binding affinity of polyamines in the pore when asparagine is present compared to serine (Fig. 9 and Supplementary Fig. 9; Supplementary Table 3). Interestingly, concurrent neutralization of the +4 aspartate to alanine ($D_{646}A$) completely abrogated this enhanced polyamine block, while having minimal impacts on ion permeation, indicating this latter residue plays a key role in polyamine bonding. This was also reported for AMPA and kainate receptors, where mutation of respective +4 aspartate and glutamate residues decreased current rectification and polyamine block while having minimal impacts on divalent ion permeation[50,65], and we corroborated these findings in this study for the human GluA2 receptor (Fig. 7). As such, our data suggests that regulation of iGluRs via the Q/R/N and +4 sites is an ancestral feature shared between AKDF and epsilon receptors. Indeed, most placozoan receptors, except those from the clades GluE2β and GluE3γ, also bear acidic aspartate or glutamate residues at the +4 site (Fig. 1b), and acidic residues are found more broadly among all types of metazoan iGluRs, including cnidarian NMDA receptors and poriferan lambda receptors[5]. Even plant iGluRs, most of which bear hydrophobic residues in the Q/R/N site (i.e., phenylalanine or tyrosine), have +4 glutamate residues[66], as does the primordial iGluR receptor from bacteria, GluR0, which bears a potassium channel-like selectivity filter but a +4 aspartate equivalent[67]. Whether conservation of this residue in non-metazoan receptors reflects even deeper origins for polyamine regulation is unclear, but it is notable that chimeric receptors comprised of kainate receptors (GluR6) bearing the pore module of a plant iGluR (i.e., AtGLR1.1 from *Arabidopsis thaliana*) conduct rectifying currents that are consistent with voltage-dependent regulation by polyamines[68]. Like resolved iGluR structures, the four $D_{646}$ residues of the heteromeric GluE1αA receptor complex are predicted to come together at the cytoplasmic entrance of the pore, well positioned to interact with the positively charged amino groups of cytoplasmic polyamines. That this ring of aspartate residues provides an electrostatically attractive entry point for polyamines to enter the pore was also indirectly supported by our perfusion experiments, where the $D_{646}A$ variant showed increased current block by 1 mM external polyamines compared to the wildtype receptor (Fig. 9f). Here, we suggest that the absence of the aspartate residues in the mutant channel prevented extracellular polyamines from traversing the pore into the cell interior, accumulating inside a vestibule within the pore and hence blocking inward cation permeation.

In sum, it seems sequence changes in the pore-loop of iGluRs occurred quite commonly, representing adaptive changes that are poised to alter $Ca^{2+}$ permeation and polyamine regulation. For GluE1αA, the emergence of a serine served to reduce both polyamine block and $Ca^{2+}$ permeation. Based on Q/R/N mutation experiments of the kainate receptor GluR6[50], the presence of a glycine in the Q/R/N site, as observed on *M. leidyi* epsilon receptors, would similarly decrease polyamine block, while a phenylalanine, present in the *B. lanceolatum* GluE1 receptor, would retain strong polyamine block. Interestingly, mammalian AMPA and kainate receptors also exhibit adaptations in polyamine regulation, in this case via complexing with corresponding ancillary subunits (reviewed in ref. 12). Although the physiological significance of variations in polyamine regulation are unclear, reduced polyamine block renders channels that are more active during bouts of excitation when the membrane is depolarized. Hence, while receptors with strong rectification are constrained to depolarize the membrane from negative resting voltages, those with reduced rectification can operate at a broader range of membrane voltages. Given their non-selective nature, this would increase the input resistance of the cell membrane to possibly dampen fast electrical impulses through conduction block.

## Materials and methods

### Identification and cloning of *T. adhaerens* iGluRs
*T. adhaerens* ionotropic glutamate receptor sequences were identified by BLAST[69] searching a whole animal mRNA transcriptome assembly[22] using a set of NMDA, AMPA and epsilon protein sequences from human and the ctenophore *Mnemiopsis leidyi* as queries. Candidate *T. adhaerens* sequences were then analyzed with SmartBlast[70] and reciprocal BLAST of the NCBI non-redundant database to confirm homology to iGluRs, InterPro[71] to predict conserved domains, and TMHMM[72] to predict transmembrane helices. This identified 13 iGluR sequences, 11 of which contained complete protein coding sequences and a minimum of three predicted transmembrane helices. To verify the mRNA sequences of these identified full-length transcripts and confirm their expression in vivo, we used nested PCR to amplify their protein coding sequences from a whole animal cDNA library utilizing primers listed in Supplementary Table 5. PCR was successful for 10 of the 11 receptors, and the corresponding DNA amplicons were cloned into the mammalian expression vector pIRES2-EGFP (Clontech) using restriction enzymes listed in Supplementary Table 5, followed by sequencing and analysis of triplicate independent clones to distinguish polymorphisms from PCR errors and determine consensus protein sequences for each receptor (submitted to NCBI with accession numbers listed in Supplementary Table 5).

### Phylogenetic analyses of iGluR protein sequences
We started by compiling a set of eukaryotic proteomes with a balanced sampling of 88 species from the clade Amorphea (including groups such as animals, choanoflagellates, and fungi) and 96 species from Diaphoreticckes (including plants and stramenopiles). Details about the included species, sources of proteomes, and their BUSCO quality metrics[73] are provided in Supplementary Table 1. To generate a species aware protein phylogeny of iGluR sequences with GeneRax[32], we first created separate species and iGluR protein phylogenies which are used as inputs for the program. For the former, single copy BUSCO genes for each species were used to build a concatenated supermatrix, by first aligning homologous gene sequences with MAFFT[74] using default parameters, then trimming these with trimAl[75] using the gappyout mode, and finally concatenation with FASconCAT[76]. The species tree was built with maximum likelihood using IQ-TREE 2[77] with the evolutionary model LG + G4 + F. Branch supports were assessed with ultrafast bootstrap 1000 replicates[78] as well as with 1000 replicates for the Shimodaira-Hasegawa-like approximate likelihood ratio test (SH-aLRT)[79]. The resulting species tree was rooted in such a way as to maintain the monophyly of the Amorphea clade and was used as a backbone for gene tree-to-species tree reconciliation (see below).

Next, we used HMMER v3.1b2[80] to generate a hidden Markov model profile of iGluRs using protein sequences from the InterPro entry

IPR001320. The HMM profiles were then used to search the noted proteomes for candidate iGluR protein sequences using an expect value (*e*-value) threshold of 1E−10, and these were then filtered to remove redundant sequences using CD-HIT[81] with a threshold of 0.95 (i.e., 95% sequence identity), except for placozoan species for which we used a threshold of 1.00 in order to retain potential transcript variants. The remaining sequences were clustered by sequence similarity using the program CLANS[82] with the alignment scoring matrix BLOSUM62 and an *e*-value threshold of 1E−14. The sequences from a high confidence iGluR cluster, defined by the stringent *e*-value 1E−65, were extracted and combined with the sequences from the InterPro iGluR entry to create a new, more refined profile with HMMER. This additional step was carried out because InterPro sequences are primarily derived from model organisms. By incorporating sequences collected from the initial HMMER search across a broad set of organisms, we sought to develop a more inclusive profile that was more likely to identify sequences in distantly related non-model organisms. A second round of HMMER with an *e*-value of 1E−10 was run with these updated profiles. For each species, CD-HIT was run with an identity threshold of 95% (and 100% for placozoan species) to reduce potential transcript variants from the second HMMER search. The resulting sequences were combined with sequences from the first round and CD-HIT was used with 100% identity in all cases to get rid of duplicates from the two rounds of HMMER. The resulting sequences were again clustered using CLANS using the same parameters as before. Connected sequences from the iGluR cluster with a *p*-value cut-off of 1E-60 (Supplementary Fig. 4 and Supplementary Data 2), were extracted. An additional filtering of the dataset was carried out to remove sequences with less than two or more than six transmembrane helices predicted with the program Phobius[83]. These extracted iGluR sequences were then aligned with MAFFT using the E-INS-i algorithm, and the alignment trimmed with trimAl in using the gappyout mode. The gene tree was inferred with maximum likelihood using IQ- TREE2, with a best-fit model according to Bayesian Information Criterion of Q.pfam+I + R10. Branch supports were assessed with 1000 ultrafast bootstrap replicates as well as 1000 SH-aLRT replicates. Any potential polytomy in the tree was randomly resolved using ETE3[84]. The resulting gene tree was used as starting tree for gene tree to species tree reconciliation using GeneRax v2.1.2[32] set to account for duplication and loss events (UndatedDL model). The utilized iGluR sequences, in raw, aligned, and aligned trimmed format, along with IQ-TREE 2 and GeneRax output files are deposited in the Zenodo repository[85].

To generate a comprehensive phylogeny exclusively of placozoan iGluRs, we manually extracted candidate protein sequences from the gene data available for four species: *T. adhaerens*[21–23], *Trichoplax* species H2[23], *Hoilungia hongkongensis*[24], and *Cladtertia collaboinventa*[25]. This was done by first BLAST searching the various databases using the *T. adhaerens* protein sequences identified in the transcriptome as queries (described above; Supplementary Table 2), followed by manual annotation as described above using SmartBlast, InterPro, and TMHMM. We excluded protein sequences with 1 or 0 predicted transmembrane helices, resulting in the identification of 15 sequences for *T. adhaerens*, (11 from the whole animal transcriptome plus 3 from the Ensembl and NCBI databases and 1 comprised of merged overlapping sequences from the transcriptome and Ensembl), 12 for *Trichoplax* species H2, 10 for *H. hongkongensis*, and 12 for *C. collaboinventa* (Supplementary Table 2). The identified were aligned with MUSCLE[86], and the alignment trimmed with trimAl[75] using a gap threshold of 0.6. This trimmed alignment was then used to infer a maximum likelihood phylogenetic tree using the program IQ-TREE2[77] with a best-fit model of Q.yeast+I + G4 identified under Bayesian Information Criterion, and 1000 ultrafast bootstrap replicates to generate node support values. Top cellular expression for the different placozoan receptors was achieved by extracting single cell RNA-Seq Umifrac values for each receptor transcript[25], and determining the average metacell value for each cell type. The placozoan iGluR sequences, in raw, aligned, and aligned trimmed format, along with IQ-TREE 2 output files are deposited in the Zenodo repository[85].

## Whole cell patch clamp electrophysiology of GluE1αA expressed in CHO-K1 cells

The cDNA of the *T. adhaerens* GluE1αA receptor was cloned into the mammalian expression vector pIRES2-EGFP, using a 5' nested primer that inserted a mammalian consensus Kozak sequence of GCCGCCACC<u>ATG</u> (Supplementary Table 5). Several mutants/variants of this receptor where also prepared via site directed mutagenesis using standard procedures and primers listed in Supplementary Table 5. For electrophysiological comparisons, we commissioned GenScript (USA) to synthesize the cDNA of the human GluA2 heterotetrameric receptor (NCBI accession number NP_001077088.2), which was cloned into the pIRES2-EGFP vector with a Kozak sequence flanking the start codon and restriction enzymes *Sal*I and *BamH*I at the 5' and 3' ends of the cDNA, respectively. Receptor cDNAs in the various pIRES2 constructs were transfected into Chinese Hamster Ovary-K1 (CHO-K1; Sigma) cells cultured in Dulbecco's Modified Eagle Medium Nutrient Mixture F-12 (DMEM/F12) supplemented with 10% Fetal Bovine Serum (FBS) and 1% penicillin-streptomycin at 37°C. Transfections were done with 2 μg of plasmid vector using the transfecting reagent PolyJet™ (SignaGen Laboratories, USA) according to the manufacturer's instructions and plated onto glass coverslips in 35 mm dishes the next day. Transfected cells were incubated at 28°C for 2 to 4 days, then coverslips were transferred to a new 60 mm dish with 5 mL of external recording solution. For the dose response, recovery from desensitization, and pharmacology experiments, the external recording solution contained: 145 mM NaCl, 4 mM KCl, 2 mM CaCl₂, 1 mM MgCl₂, 10 mM HEPES (pH 7.4 with NaOH; 313 mOsm/Kg). The internal solution contained: 145 mM CsCl, 2.5 mM NaCl, 1 mM EGTA, 4 mM Mg-ATP, 10 mM HEPES (pH 7.2 with CsOH; osmolality adjusted to 304 mOsm/Kg with D-glucose). For the ion selectivity experiments, the external solution contained: 150 mM XCl ion (where X = Na, K, Li, or Cs), 10 mM tetraethylammonium chloride (TEA-Cl), and 10 mM HEPES (pH 7.4 with XOH), and the internal solution contained: 150 mM NaCl, 10 mM EGTA, 10 mM TEA-Cl, and 10 mM HEPES, 4 Mg-ATP (pH 7.2 with NaOH). For calcium selectivity experiments, the internal solution consisted of 100 mM XCl, 10 mM EGTA, 10 mM TEA-Cl, 10 mM HEPES, and 4 mM Mg-ATP (pH 7.2 with XOH; osmolality adjusted to 326 mOsm/Kg). The external solution contained: 4 mM CaCl₂, 155 mM TEA-Cl, 10 mM HEPES (pH 7.4 with TEA-OH; osmolarity adjusted to 326 mOsm/Kg). Agonists and pharmacological compounds used in this study were dissolved in corresponding external solutions. Whole-cell macroscopic currents were recorded using an Axopatch 200B amplifier coupled to a Digidata 1550 A digitizer, using the pClamp 11 software (Molecular Devices, USA). Patch pipettes were pulled using a P-1000 micropipette puller (Sutter, USA), from thick-walled borosilicate tubing (1.5 and 0.86 outer and inner diameter, respectively), to a resistance of between 1 and 6 megaohms. $EC_{50}$ and $IC_{50}$ values for dose-response curves were determined by fitting monophasic dose-response curves over the data using the software package Origin 2017 (OriginLab) and the equation below.

$$y = \frac{1}{1 + 10^{(\log_{10} x_0 - x)p}}$$

Where y is normalized current, x is the $\log_{10}$(ligand concentration), $x_0$ is the $EC_{50}$ and p is the Hill slope. The dual CNQX $EC_{50}$ and $IC_{50}$ values reported for m3 variant of GluE1αA were obtained by fitting a biphasic dose response curve with one stimulatory and one inhibitory phase over the data using the automated curve fitting software Dr-fit[87].

Time constant for the recovery from desensitization was calculated by fitting mono-exponential curves over the data using Origin with the following equation:

$$y = 1 + Ae^{x/\tau}$$

Where y is the peak current normalized to pre-pulse, A is the amplitude, x is the interval between pre-pulse and test pulse and τ is the time constant.

The relative permeability pX⁺/pNa⁺ determined for TGluE1αA, where X⁺ is a monovalent cation (Na⁺, Li⁺, Cs⁺, or K⁺), was determined using the biionic equation[88]:

$$P_X/P_{Na} = e^{\frac{F(E_{rev,X} - E_{rev,Na})}{RT}}$$

Where F is the Faraday's constant (F = 96485.33 C/mol), R is the gas constant (R = 8.314463 J/K· mol), T is the ambient temperature (T = 294 K), and $E_{rev}$ is the reversal potential.

The relative permeability ratio pCa²⁺/pX⁺ was determined using the following monovalent/divalent biionic equation[89]:

$$pCa^{2+}/pX^+ = \left(\frac{[X^+]_{in}}{4[Ca^{2+}]_{out}} e^{\frac{E_{rev}F}{RT}}\right)\left(e^{\frac{E_{rev}F}{RT}} - 1\right)$$

Where $[X^+]_{in}$ is the concentration of the internal monovalent cation and $[Ca^{2+}]_{out}$ is the concentration of calcium ions in the external solution.

### Macroscopic outside-out patch clamp electrophysiology of GluE1αA and GluA2 expressed in HEK-293T/17 cells

HEK-293T/17 cells (ATCC) were maintained in MEM supplemented with GlutaMAX and 10% fetal bovine serum (FBS) at 37 °C and 5% $CO_2$. Cells were plated at low density (1.6 × 104 cell/ml) on poly-D-lysine-coated 35-mm dishes. Transient transfections were carried out 48 h post-plating using the calcium phosphate precipitation method. After 6–8 hours, cells were washed twice with PBS and maintained in fresh medium. Cells transfected with mammalian GluA2 cDNA (plasmid vector described above) were returned to an incubator at 37 °C for 24–48 hours and cells transfected with Epsilon GluE1αA were incubated at 28 °C for 48–72 hours. In cells co-transfected with TARP-γ2, 30 μM NBQX was included in the medium to minimize AMPAR/auxiliary-induced cytotoxicity. All constructs were co-transfected with a plasmid encoding enhanced green fluorescent protein (eGFP) to identify transfected cells. Electrophysiology responses were recorded from outside-out patches excised from transfected cells. Recording pipettes were composed of borosilicate glass (3 to 6 megaohm, King Precision Glass) coated with dental wax to reduce electrical noise. All recordings were performed using Axopatch 200B amplifier (Molecular Devices, LLC). Current records were filtered at 5 or 10 kHz and sampled at 25–50 kHz. Series resistance (3 to 12 megaohms) was compensated by 95%. Recordings were performed at a range of holding potentials from −100 mV to 100 mV to study polyamine block. Data was acquired using pClamp9 software (Molecular Devices, LLC). All experiments were performed at room temperature. All chemicals were purchased from Sigma-Aldrich, unless otherwise indicated. External solution contained (in mM): 150 NaCl, 5 HEPES, 0.1 $CaCl_2$, 0.1 $MgCl$, and 2% phenol red, pH 7.3–7.4. Internal solution contained (in mM): 115 NaCl, 10 NaF, 5 HEPES, 5 $Na_4$BAPTA (Life Technologies), 0.5 $CaCl_2$, 1 $MgCl_2$, and 0.03 Spm. The osmotic pressure of all solutions was adjusted to 295 to 300 mOsm with sucrose. Concentrated (10x) agonist stock solutions were prepared by dissolving ether L-glutamate or glycine in the appropriate external solution and adjusting the pH to 7.3 to 7.4 and were stored at −20 °C. Stocks were thawed on the day of the experiment and used to prepared agonist-containing external solutions. Raw data for all electrophysiology experiments are provided in Supplementary Data 3. Current-voltage (I-V) relationships were fit using Origin 7 (OriginLab) using a ninth order polynomial function to estimate the reversal potential ($V_{rev}$).

Agonist-evoked membrane conductance (G) was calculated using the following equation:

$$G = \frac{I}{(V - V_{rev})}$$

Conductance-voltage (G-V) relationships were fit using Origin 7 (OriginLab) with the single permeant blocker model as done previously[46]:

$$G = \frac{G_{max}}{1 + \frac{[Spm]}{K_d}}$$

Where $G_{max}$ is the maximal conductance, $[Spm]$ is the internal Spm concentration, and $K_d$ is the dissociation constant.

$K_d$ is defined as:

$$K_d = \frac{k_{off} + k_{perm}}{k_{on}} = \frac{sum\ of\ exit\ rates}{binding\ rate}$$

And redefined as:

$$K_d = g * \exp(V/h) + L * \exp(V/k)$$

Where h and k are voltage dependencies of g and L, respectively, and at 0 mV:

$$g = \frac{k_{off}}{k_{on}} and\ L = \frac{k_{perm}}{k_{on}}$$

GluE1αA $D_{646}$A intrinsic G-V relationships were fit with the following exponential equation using Origin 7:

$$G = \left(1 + (G_0 - 1) \exp\left(\frac{V}{V_c}\right)\right)$$

Where $G_0$ is the minimal conductance, and $V_c$ is a voltage constant.

### Structural modelling and docking

The structure of the wildtype GluE1αA receptor, as well as various mutants in the ligand binding and pore domains, were predicted with AlphaFold2[38]. For spermine docking experiments, predicted pore domains were opened in silico as described previously[90]. Details of this method adapted for the current study are provided in the results section. Additionally, the staring conformation of the ring of engineered asparagines in models of the $S_{642}$N variant of GluE1αA was biased by imposed inter-subunit hydrogen bonds[91], where each asparagine donated two bonds. This bias ensured orientation of the sidechain oxygen atoms towards the pore axis. It should be noted that this orientation was imposed only once to the apo-channel model, but in the spermine-channel model, sidechain asparagines were free to move during energy minimizations. Energy optimizations and ligand docking into the GluE1αA receptor were performed with the program ZMM as described in a recent study[92]. Briefly, energy was calculated with the AMBER force field[93,94], distance- and environment-dependent dielectric function[95] and implicit solvent[96]. The energy was optimized by Monte Carlo (MC) energy minimizations[97]. During MC minimizations, we used "pin" constraints to ensure the similarity of the backbone conformation in the model and the starting structure. A pin is a flat-bottom parabolic energy function that imposes energy penalties if an alpha carbon atom in the model deviates by more than 1 Å from the starting position and imposes an energy penalty with the force constant of 10 kcal·mol⁻¹·Å⁻². Computations were performed using facilities provided by Compute Ontario (https://www.computeontario.ca/) and the Digital Research Alliance of Canada (https://www.alliancecan.ca).

### Statistics and reproducibility

Details on statistics can be found in the figure legends. Transfections were performed at least two independent times. For simple pairwise comparisons, unpaired two-tailed Student's t tests were conducted as indicated. For datasets involving comparisons between multiple groups, one-way and two-

way ANOVAs with Tukey's HSD were completed as indicated. Statistical analyses were conducted using Origin 7 (OriginLab) and custom statistical software generously provided by Joe Rochford (McGill University). Significance level was set at 0.05 and is denoted as *p* values as indicated in the figure legends. All sample sizes are indicated in the figure legends.

## Reporting summary

Further information on research design is available in the Nature Portfolio Reporting Summary linked to this article.

## Data availability

All presented data are available in the text, Supplementary Figs., and Supplementary Data files, and source data for all plots are provided in Supplementary Data 3. Data sets used for the phylogenetic analyses presented in Fig. 1 and Supplementary Figs. 1 to 4 are accessible through a Zendo repository[85].

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

## Acknowledgements
This research was funded by an NSERC Discovery Grant (RGPIN-2021-03557), an NSERC Discovery Accelerator Supplement (RGPAS-2021-00002), an Ontario Early Researcher Award (ER17-13-247), and a Canadian Foundation for Innovation Grant (35297) to A. Senatore, an Ontario Graduate Scholarship to A. Singh, a Natural Sciences and Engineering Research Council of Canada (RGPIN-2020-07100) and ongoing funding of the Sechenov Institute to B.S.Z., a CIHR operating grant (FRN 191714) to D. B., and a BBSRC Fellowship (BB/W010305/2) and Royal Society funding (RG \R1\241397) to L.A.Y.G. We thank Denis Tikhonov for helpful discussions.

## Author contributions
A. Senatore conceived the project; C. D. Yanartas, A. Singh, and A. Senatore conducted the initial molecular biology experiments to clone and sequence the *T.adhaerens* iGluRs from a cDNA library. A. Singh and A. Senatore designed the ligand specificity, ion selectivity, and pharmacology experiments, and these were carried out by A. Singh. L.A. Yanez-Guerra, A. Aleotti, and A. Senatore designed and carried out the phylogenetic analyses. B.S. Zhorov, A. Singh, and A. Senatore designed the structural modeling and ligand/polyamine docking experiments that were done by B.S. Zhorov with contributions from A. Senatore and A. Singh. A. Senatore, A. Singh, F. Miguez-Cabello, D. Bowie, C.C. Koens, and Y. Song designed the polyamine electrophysiology experiments that were done by A. Singh, C.C. Koens, Y. Song, and F. Miguez-Cabello; All authors contributed to the writing and revision of the manuscript.

## Competing interests
The authors declare no competing interests.
