## [Transparent Peer Review file · Communications Biology]

Evolution of iGluR ligand specificity, polyamine regulation, and ion selectivity inferred from a placozoan Epsilon receptor

Corresponding Author: Dr Adriano Senatore

Version 0:

Reviewer comments:

Reviewer #1

(Remarks to the Author)
Review of Sing et al.,

This is an interesting study delving into uncharacterized iGluRs from evolutionarily distant organisms. The functional characterization is well done and useful to the field. And the combination of sequence and species-rooted trees will be a valuable addition to this growing area of interest. I have only one major issue and several minor concerns aimed at improving clarity and rigor.

CNQX experiments are not well designed nor represented. It would be better to pre-apply CNQX and then jump into glu or gly. Instead, the authors have co-applied CNQX and the agonist, setting up a "race" experiment as in Jones et al., J Neurosci 1998 Figure 4. Results from such experiments are driven by the ratio of macroscopic binding rates between ligands, and have less information about ligand dissociation. Since affinity (KD) is dissociation(kd)/association(ka) and this experiment primarily reflects association, it is incorrect to refer to the IC50s as measurements of affinity. This experiment also underrepresents the inhibitory potency of low concentrations of antagonist. 10 uM CNQX may well block 3 mM glycine more effectively at wt GluE1alphaA receptors (Figure 4C) if it was pre-applied and allowed to equilibrate. A better experiment is to pre-apply various concentrations of CNQX and jump into agonist, allowing equilibrium to occur and then measuring current. The distinction is important if one wants to say anything about affinity. The authors should either conduct the pre-equilibration experiment or alter their text to reflect what they have actually measured (i.e. an estimate of differences in macroscopic binding rates and not affinity). Regardless, the figures give the impression that the authors are continually applying agonist and jumping into antagonist. Please re-draw the ligand application bars to more accurately reflect what's actually happening.

Minor

Figure 3C, what level of statistical significance do the a, b, and c denote?

The voltage ranges for all constructs (Figure 5A) share a similar color palette as that used for each construct across all voltages (Figure 5B-E). This is needlessly confusing. Better to just use black or grey scale for the voltages since the audience probably doesn't need help discerning which current trace comes from which voltage, and use color for the various channels tested. Similar issue in Figure 8.

Figure 6, consider putting the construct name at the top of the first rows of traces in larger letters. The current placement as small text in the second row is harder to see.

The Kd equation for polyamine block in the methods section has koff, kperm and kon as parameters derived from fitting the GV curves. But Koff and kperm are dependent on the specific value of kon, which was not measured independently. Was Kon fixed? If so, what was the value and justification?

Reviewer #2

(Remarks to the Author)
Summary:

The manuscript by Singh et al. is a very nice study to understand the evolution of Epsilon-type iGluR, from the aspect of ligand specificity and ion channel pore properties. The authors generated a phylogeny of eukaryotic iGluRs and a focused analysis of iGluR homologues from the early-branching invertebrate phylum Placozoa. The authors characterized the biophysical functions of an Epsilon receptor, GluE1 α A, from the model placozoan species *Trichoplax adhaerens*. The authors found that changing just three amino acids could change the ligand specificity from glycine to glutamate. The authors further found that an atypical serine residue in the channel pore diminished Ca²⁺ permeability and polyamine block compared to typical iGluRs. Furthermore, the authors found that a conserved aspartate downstream of that serine residue is also crucial for polyamine block. The authors demonstrated the conservation of polyamine block between AKDF and Epsilon receptors.

The manuscript is well-written and clear, the figures are mostly very clear, and the data is very convincing. Below are some detailed comments and questions.

1. In Results "The *Trichoplax* GluE1 α A receptor has a broad ligand specificity and moderately fast recovery from desensitization", can you clarify the EC50 value for glycine.
2. In Results "Mutation of key residues in the ligand binding domain switches ligand specificity from glycine to glutamate", it seems that the m2 and m3 mutants still had some activity in response to Gly/D-Ser: I wonder how they affected the EC50 for Gly/D-Ser and if, they affect Gly affinity or channel open probability and efficacy. Do you possibly know how F732Y single mutation affects the Ala/Gly EC50?
3. In Results "Mutation of key residues in the ligand binding domain switches ligand specificity from glycine to glutamate", you talked about the how glycine binds to the receptor using dockings. "Glycine binding to the wildtype channel is stabilized mainly by salt bridges with residues D705 and R485 and non-bonded attraction to K728 and Y450", however, I couldn't see the residue K728 in figure 3f-i. I think it should be showed in your figures since you talk about this residue.
4. In Results "Mutations that alter ligand specificity also affect sensitivity to vertebrate iGluR agonists and the AMPA/Kainate blocker CNQX", the m2 mutant exhibited a biphasic response but the m3 mutant did not: here the lowest concentration tested is 0.1 μ M, I wonder if you tested even lower concentrations, like 1 nM or 10 nM, could you also see the biphasic response by CNQX?
5. In Results "A unique serine residue in the Q/R/N site decreases voltage-dependent regulation by polyamines", the I-V figures showed many mutants in one graph, which makes it difficult to distinguish the results, maybe you can use more contrasted symbols for different mutants.
6. In Results "A unique serine residue in the Q/R/N site decreases voltage-dependent regulation by polyamines", I think there is a typing error, "GluE1 α A Q642R experiments were unable to be replicated in outside-out patches due to poor expression", is it the mutant S642R?
7. In Results "The Q/R/N serine also diminishes Ca²⁺ permeation", you said you observed only minimal change imposed by the D646A mutation. But compared the single mutant S642Q, the double mutant S642Q and D646A showed more than 50% decrease in I-100mV/I+80mV from figure 8 b-e, is this not a big influence on Ca²⁺ permeability?
8. "By mutating just three amino acids in the ligand binding domain, we were able to completely switch the ligand specificity of GluE1 α A from glycine/serine to glutamate, providing experimental support for the hypothesis that just a small set of residues in the ligand binding domain determine ligand specificity." I want to ask if you think these determinants of ligand specificity are common to all iGluRs or just this branch of the Epsilon family, as we know that e.g. Ctenophore Epsilon (Alberstein et al., 2015) have very different residues at these positions yet bind glycine.

Version 1:

Reviewer comments:

Reviewer #1

(Remarks to the Author)

The authors have satisfied nearly all my concerns, except for the letters denoting significance in Figure 3C and 4B. I understand that the letters indicate something is significant and the F/p values are in the legend. But my issue is what the actual letters stand for. Does "a" mean $p < 0.05$ and $b < 0.01$ for example? Why have a, b, c etc. at all if the specific meanings are never explained.

Also, in the abstract lines about CNQX, the authors use the phrase "the drug blocker CNQX." Perhaps this was a typo and they meant either drug or blocker?

Reviewer #2

(Remarks to the Author)

I appreciate that the authors have thoroughly addressed most of my previous comments. The revisions have significantly enhanced the quality of this excellent study.

1. The experiments they did with m1 variant in Fig3 clearly showed the contribution of mutation F732Y to ligand specificity.
2. The new CNQX pharmacology experiments improved the analysis of CNQX sensitivity.
3. The modifications of adding Ctenophore glycine-sensitive iGluRs are helpful to understand the molecular determinants of glycine specificity, but they don't clarify my point. Reporting just the one Ctenophore Epsilon result is just one result – and arguably a unique one, as the authors write. The authors could discuss various other Gly-binding iGluR subunits – epsilon, AKDF, NMDA – and what amino acid residues they have at these positions. Is mutating the same three amino acids likely to switch the ligand specificity in these receptors too?

Overall, the paper is clear, comprehensive, and ready for publication perhaps with the one minor adjustment above. I do not need to see the manuscript again before publishing.

We would like to thank the reviewers for their time reviewing the manuscript, and for their insightful feedback. In this revised manuscript, we hope to have addressed all their concerns, including additional experiments that we conducted. Altogether, we feel the manuscript has been significantly improved.

Please note that we also made a few small edits to the abstract, and small corrections throughout the manuscript (e.g., an incorrect reference, small typographical errors), which can be viewed in the marked-up version.

Reviewer #1 (Remarks to the Author):

Review of Sing et al.,

This is an interesting study delving into uncharacterized iGluRs from evolutionarily distant organisms. The functional characterization is well done and useful to the field. And the combination of sequence and species-rooted trees will be a valuable addition to this growing area of interest. I have only one major issue and several minor concerns aimed at improving clarity and rigor.

CNQX experiments are not well designed nor represented. It would be better to pre-apply CNQX and then jump into glu or gly. Instead, the authors have co-applied CNQX and the agonist, setting up a “race” experiment as in Jones et al., J Neurosci 1998 Figure 4. Results from such experiments are driven by the ratio of macroscopic binding rates between ligands, and have less information about ligand dissociation. Since affinity (KD) is dissociation(kd)/association(ka) and this experiment primarily reflects association, it is incorrect to refer to the IC50s as measurements of affinity. This experiment also underrepresents the inhibitory potency of low concentrations of antagonist. 10 uM CNQX may well block 3 mM glycine more effectively at wt GluE1alphaA receptors (Figure 4C) if it was pre-applied and allowed to equilibrate. A better experiment is to pre-apply various concentrations of CNQX and jump into agonist, allowing equilibrium to occur and then measuring current. The distinction is important if one wants to say anything about affinity. The authors should either conduct the pre-equilibration experiment or alter their text to reflect what they have actually measured (i.e. an estimate of differences in macroscopic binding rates and not affinity). Regardless, the figures give the impression that the authors are continually applying agonist and jumping into antagonist. Please re-draw the ligand application bars to more accurately reflect what’s actually happening.

Thank you for noting this important caveat about our CNQX experiments. In response, we have done new pharmacology experiments using the suggested blocker pre-application strategy, perfusing CNQX at different concentrations alone for a period of 2 seconds, then for 1 additional second co-applying the same concentration of CNQX

along with saturating concentrations of 10 mM glycine for the wildtype receptor, or 10 mM glutamate for the m3 variant. We decided to not do additional experiments for the m2 variant, for several reasons. Primarily, we feel the pharmacology is not a central focus of this current study and should be done in greater detail in a separate focused study, exploring differences in agonist and antagonist pharmacology of the different receptor variants in a comprehensive manner. Changes in the manuscript to address these new experiments are in lines 384-394 (results) and 898-992 (methods) in the marked-up version of the revised manuscript.

Minor

Figure 3C, what level of statistical significance do the a, b, and c denote?

This information is provided in the corresponding figure legend (lines 1277-1281):
“Letters above the bars denote statistically significant differences among ligand types for post hoc Tukey tests ($p < 0.05$) after One Way ANOVAs (glycine: $F=295.00$, $p=1.11E-16$; glutamate: $F=336.17$, $p=1.11E-16$; D-serine: $F=209.73$, $p=2.78E-15$; L-serine: $F=534.65$, $p=0$; alanine: $F=279.38$, $p=2.78E-15$; valine: $F=20.77$, $p=2.35E-6$).”

The voltage ranges for all constructs (Figure 5A) share a similar color palette as that used for each construct across all voltages (Figure 5B-E). This is needlessly confusing. Better to just use black or grey scale for the voltages since the audience probably doesn't need help discerning which current trace comes from which voltage, and use color for the various channels tested. Similar issue in Figure 8.

We have changed the colors as suggested.

Figure 6, consider putting the construct name at the top of the first rows of traces in larger letters. The current placement as small text in the second row is harder to see. The Kd equation for polyamine block in the methods section has koff, kperm and kon as parameters derived from fitting the GV curves. But Koff and kperm are dependent on the specific value of kon, which was not measured independently. Was Kon fixed? If so, what was the value and justification?

The apparent affinity of polyamine block determined by the Kd equation was governed by the sum of the terms g and L, which correspond to koff/kon and kperm/kon, respectively, as originally described in the Methods section of Bowie et al., 1998 Journal of Neuroscience, page 8176. As such, we did not assume a specific value for kon in our estimation of Kd. However, as noted in Bowie et al., 1998 and subsequent papers from the Bowie lab, the primary source of voltage dependency of polyamine block is due to the rate of unblock (i.e. koff) and/or the rate of permeation (i.e. kperm). We have added a line to the methods section of the manuscript to add further clarification (lines 1050 and 1051; highlighted in yellow below):

K_d is defined as:

$$K_d = \frac{k_{off} + k_{perm}}{k_{on}} = \frac{\text{sum of exit rates}}{\text{binding rate}}$$

And redefined as:

$$K_d = g * \exp(V/h) + L * \exp(V/k)$$

Where h and k are voltage dependencies of g and L , respectively, and at 0 mV:

$$g = \frac{k_{off}}{k_{on}} \text{ and } L = \frac{k_{perm}}{k_{on}}$$

Reviewer #2 (Remarks to the Author):

Summary:

The manuscript by Singh et al. is a very nice study to understand the evolution of Epsilon-type iGluR, from the aspect of ligand specificity and ion channel pore properties. The authors generated a phylogeny of eukaryotic iGluRs and a focused analysis of iGluR homologues from the early-branching invertebrate phylum Placozoa. The authors characterized the biophysical functions of an Epsilon receptor, GluE1 α A, from the model placozoan species *Trichoplax adhaerens*. The authors found that changing just three amino acids could change the ligand specificity from glycine to glutamate. The authors further found that an atypical serine residue in the channel pore diminished Ca²⁺ permeability and polyamine block compared to typical iGluRs. Furthermore, the authors found that a conserved aspartate downstream of that serine residue is also crucial for polyamine block. The authors demonstrated the conservation of polyamine block between AKDF and Epsilon receptors.

The manuscript is well-written and clear, the figures are mostly very clear, and the data is very convincing. Below are some detailed comments and questions.

1. In Results “The *Trichoplax* GluE1 α A receptor has a broad ligand specificity and moderately fast recovery from desensitization”, can you clarify the EC₅₀ value for glycine.

Thank you for noticing this inadvertent omission. We have added the EC₅₀ value to the text in the results (lines 297-301; highlighted below):

“Dose-response curve analysis of activating ligands revealed strongest sensitivity to alanine, followed by glycine, L-serine, D-serine, all with EC₅₀ values in the sub-millimolar range (*i.e.*, L-alanine = 0.049 ±0.011 mM, glycine = 0.11 ±0.02 mM, L-serine = 0.13 ±0.02 mM, and D-serine = 0.19 ±0.04 mM), followed by valine with an EC₅₀ of 1.51 ±0.24 mM (Figure 2b and c, Supplementary Figure 5).”

2. In Results “Mutation of key residues in the ligand binding domain switches ligand specificity from glycine to glutamate”, it seems that the m2 and m3 mutants still had some activity in response to Gly/D-Ser: I wonder how they affected the EC50 for Gly/D-Ser and if, they affect Gly affinity or channel open probability and efficacy. Do you possibly know how F732Y single mutation affects the Ala/Gly EC50?

We agree with this suggestion and hence carried out additional experiments on a single F732Y variant (added to new Figure 3).

We also added to/revised the following sections to address this new data:

Results (lines 320-342):

“...we mutated the serine and isoleucine residues at positions 653 and 655 of GluE1 α A to glycine and threonine, respectively, to convert its predicted ligand specificity from glycine to glutamate (named the m2 variant, with the mutations S₆₅₃G and I₆₅₅T) (Figure 3a). We also created a triple mutant, with an additional phenylalanine to tyrosine mutation at position 732 (F₇₃₂Y; m3 variant), to make the *T. adhaerens* epsilon receptor resemble GluN2A and GluA2 receptors from human, reasoning that this mutation might enhance glutamate sensitivity. We also generated a single F₇₃₂Y mutant (m1 variant), to assess the contribution of this amino acid to ligand specificity in isolation.

Application of various activating ligands at 10 mM concentrations and normalizing peak currents to either glycine (wildtype and m1), or glutamate (m2 and m3), revealed that the m2 variant of GluE1 α A became sensitive to glutamate, while becoming proportionately less sensitive to glycine, L-serine, D-serine, alanine, and valine (Figure 3b and c). The triple mutant (m3), bearing the additional F₇₃₂Y mutation, almost completely lost sensitivity to the tested non-glutamate ligands, in lieu of robust glutamate-activated currents. On its own, the F₇₃₂Y mutation did not diminish the relative sensitivity to glycine nor cause glutamate sensitivity, instead causing decreased sensitivity to D-serine, increased sensitivity to L-serine and alanine, and completely attenuated sensitivity to valine. Dose response analysis revealed that the single F₇₃₂Y mutant had decreased sensitivity to glycine compared to wildtype (*i.e.*, glycine EC50 = 0.11 \pm 0.02 mM for wildtype vs. 0.30 \pm 0.04 for m1), while it increased sensitivity to glutamate when created alongside the S₆₅₃G and I₆₅₅T mutations (*i.e.*, glutamate EC50 = 1.81 \pm 0.20 mM for m2 vs. 0.61 \pm 0.06 for m3) (Figure 3d).”

Discussion (lines 739-746):

“In addition, the F₇₃₂Y mutation introduces a hydroxyl group into the ligand binding pocket, increasing its polarity and perhaps helping to attract the side chain carboxyl group of glutamate into the pocket. This latter mutation would also diminish the hydrophobicity of the pocket, perhaps decreasing binding affinity for hydrophobic amino acid ligands, consistent with the complete loss of valine sensitivity by the F₇₃₂Y variant

of GluE1 α A tested at 10 mM (Figure 3b and c). However, this same variant also became slightly more sensitive to alanine, relative to the wildtype receptor, indicating more nuanced factors contribute to receptor activation by hydrophobic amino acids.”

3. In Results “Mutation of key residues in the ligand binding domain switches ligand specificity from glycine to glutamate”, you talked about the how glycine binds to the receptor using dockings. “Glycine binding to the wildtype channel is stabilized mainly by salt bridges with residues D705 and R485 and non-bonded attraction to K728 and Y450”, however, I couldn’t see the residue K728 in figure 3f-i. I think it should be showed in your figures since you talk about this residue.

We have added the K728 residue to the panels figure 3f to i.

4. In Results “Mutations that alter ligand specificity also affect sensitivity to vertebrate iGluR agonists and the AMPA/Kainate blocker CNQX”, the m2 mutant exhibited a biphasic response but the m3 mutant did not: here the lowest concentration tested is 0.1 μ M, I wonder if you tested even lower concentrations, like 1 nM or 10 nM, could you also see the biphasic response by CNQX?

Please see our response to reviewer 1 above, where we describe new CNQX pharmacology experiments. We suggest that, while more detailed pharmacology experiments would be very interesting, these would be more appropriate for a separate study. In short, we feel that pharmacology is not a central focus of this current study which instead focuses on characterizing the ligand specificity and ion permeation properties of the placozoan GluE1 α A epsilon receptor. Nonetheless, as astutely implied by the reviewer, our new experiments demonstrate that the m3 variant does indeed did show agonist activity from CNQX, at 1 μ M, when the drug is pre-applied prior to the ligand. This contrasts our previous experiments where this was not the case for this variant at this same concentration, likely because the drug and ligand were applied simultaneously. Changes in the manuscript accounting for these new experiments are in lines 384-392 (results) and 1328-1346 (figure legend).

5. In Results “A unique serine residue in the Q/R/N site decreases voltage-dependent regulation by polyamines”, the I-V figures showed many mutants in one graph, which makes it difficult to distinguish the results, maybe you can use more contrasted symbols for different mutants.

As suggested, we have changed the plots in Figures 5, S7, 8, and S8 to make the symbols more distinguishable. We also removed the colors of the current traces and voltage step protocol in Figures 5 And 8 to prevent confusion, as requested by reviewer 1.

6. In Results “A unique serine residue in the Q/R/N site decreases voltage-dependent regulation by polyamines”, I think there is a typing error, “GluE1 α A Q642R experiments

were unable to be replicated in outside-out patches due to poor expression”, is it the mutant S642R?

Yes, this was a mistake. Thank you for pointing it out. We have made the correction (line 497).

7. In Results “The Q/R/N serine also diminishes Ca²⁺ permeation”, you said you observed only minimal change imposed by the D646A mutation. But compared the single mutant S642Q, the double mutant S642Q and D646A showed more than 50% decrease in I-100mV/I+80mV from figure 8 b-e, is this not a big influence on Ca²⁺ permeability?

We agree with this suggestion, and have rewritten the results section to provide a clearer and more accurate description of this data (lines 578-595):

“Lastly, and consistent with our monovalent vs. monovalent experiments, the D_{646A} variant of GluE1 α A produced more linear I-V curves compared to wildtype, attributable to loss of voltage-dependent polyamine block (Figure 8b to e), also apparent in the more linear G-V plots for this variant (Figure 8g and Supplementary Figure 8b to e). This mutation also produced small but statistically significant changes in calcium vs. monovalent permeation, with lower pCa²⁺/pX⁺ values compared to wildtype for all bi-ionic conditions, respectively, with 0.81 \pm 0.14 vs. 0.61 \pm 0.07 for Na⁺_{in}:Ca²⁺_{out}, 1.01 \pm 0.28 vs. 0.59 \pm 0.05 for Li⁺_{in}:Ca²⁺_{out}, 0.67 \pm 0.11 vs. 0.61 \pm 0.04 for K⁺_{in}:Ca²⁺_{out}, and 1.14 \pm 0.10 vs. 0.99 \pm 0.06 for Cs⁺_{in}:Ca²⁺_{out} (Figure 8f and Supplementary Table 4b). Combined with the S_{642Q} mutation, the D_{646A} mutation had no effects in the presence of Na⁺ and K⁺ relative to S_{642Q} on its own, but did reduce Ca²⁺ permeability in the presence of the non-physiological ions Li⁺ (*i.e.*, 10.79 \pm 0.54 for S_{642Q} vs. 7.08 \pm 0.54 for S_{642Q} plus D_{646A}) and Cs⁺ (7.85 \pm 0.56 for S_{642Q} vs. 6.46 \pm 1.14 for S_{642Q} plus D_{646A}) (Figure 8f and Supplementary Table 4b). Altogether, these experiments corroborate our previous observations that the presence of a serine in the Q/R/N site diminishes polyamine block, and in addition, reduces Ca²⁺ permeation.”

8. “By mutating just three amino acids in the ligand binding domain, we were able to completely switch the ligand specificity of GluE1 α A from glycine/serine to glutamate, providing experimental support for the hypothesis that just a small set of residues in the ligand binding domain determine ligand specificity.” I want to ask if you think these determinants of ligand specificity are common to all iGluRs or just this branch of the Epsilon family, as we know that e.g. Ctenophore Epsilon (Alberstein et al., 2015) have very different residues at these positions yet bind glycine.

Thank you, this is an important point. We have added the following to the discussion to address this (lines 774-780):

“Lastly, it is worth noting that epsilon receptors from ctenophores evolved a unique mechanism for glycine specificity compared to other metazoan iGluRs, including epsilon receptors from other species. Indeed, ctenophore epsilon subunits have different residues at positions 653, 655, and 704 than those associated with glycine specificity [11], but still bind glycine [8, 9]. Crystal structures of ML032222a and PbiGluR3 from the respective ctenophore species *M. leidyi* and *P. bachei* revealed a salt bridge between the two lobes of the LBD that traps glycine in the binding pocket, rendering a unique mechanism for glycine binding [8, 9].”

We would like to thank the reviewers again for their feedback. We have made the requested revisions, please see below.

Reviewers' comments:

Reviewer #1 (Remarks to the Author):

The authors have satisfied nearly all my concerns, except for the letters denoting significance in Figure 3C and 4B. I understand that the letters indicate something is significant and the F/p values are in the legend. But my issue is what the actual letters stand for. Does "a" mean $p < 0.05$ and $b < 0.01$ for example? Why have a, b, c etc. at all if the specific meanings are never explained.

We have revised the two figure legends by adding an explanation that we hope resolves this issue:

Figure 3 legend (lines 1250 to 1254):

“Letters above the bars denote statistically significant differences for ligand type based on post hoc Tukey tests ($p < 0.05$) after One Way ANOVAs (glycine: $F=290.12$, $p=1.11E-16$; glutamate: $F=336.17$, $p=1.11E-16$; D-serine: $F=209.73$, $p=2.78E-15$; L-serine: $F=534.65$, $p=0$; alanine: $F=279.39$, $p=2.78E-15$; valine: $F=20.76$, $p=2.35E-6$). Specifically, means/bars with the same letters above them are not statistically different, while those with different letters are different based on Tukey tests.”

Figure 4 legend (lines 1299 to 1303):

“Letters above the bars denote statistically significant differences for applied compounds based on post hoc Tukey tests ($p < 1E-4$) after One Way ANOVAs (glycine: $F=809.23$, $p=5.55E-16$; glutamate: $F=367.97$, $p=1.79E-13$; AMPA: $F=126.50$, $p=4.07E-10$). Specifically, means/bars with the same letters above them are not statistically different, while those with different letters are different based on Tukey tests.”

Also, in the abstract lines about CNQX, the authors use the phrase “the drug blocker CNQX.” Perhaps this was a typo and they meant either drug or blocker?

We have removed the word “drug” from the abstract (line 39). Thank you for noting this mistake.

Reviewer #2 (Remarks to the Author):

I appreciate that the authors have thoroughly addressed most of my previous comments. The revisions have significantly enhanced the quality of this excellent study.

1. The experiments they did with m1 variant in Fig3 clearly showed the contribution of mutation F732Y to ligand specificity.

Yes, thank you for this suggestion.

2. The new CNQX pharmacology experiments improved the analysis of CNQX sensitivity.

Thanks to both reviewers for this suggestion.

3. The modifications of adding Ctenophore glycine-sensitive iGluRs are helpful to understand the molecular determinants of glycine specificity, but they don't clarify my point. Reporting just the one Ctenophore Epsilon result is just one result – and arguably a unique one, as the authors write. The authors could discuss various other Gly-binding iGluR subunits – epsilon, AKDF, NMDA – and what amino acid residues they have at these positions. Is mutating the same three amino acids likely to switch the ligand specificity in these receptors too?

We appreciate this clarification, and we agree that we missed an opportunity to provide more background and highlight the relevance of our work in a broader context. We have therefore added to and modified our discussion to address this concern:

The following sentence was added to the start of the discussion section entitled "Insights into iGluR ligand-specificity." (lines 686 to 688):

"Ligand specificity of iGluRs has been subject to considerable study, spurred by early realizations that different types of receptor subunits respond differently to natural ligands (reviewed in [3])."

We then added an entire new paragraph to elaborate on the prediction of iGluR ligand-specificity (lines 706 to 716):

"As noted in the results, Ramos-Vicente *et al.* [11] reviewed known structural determinants for ligand-binding by various iGluRs to propose that five key amino acids within the ligand binding domain are predictive for glutamate vs. glycine/D-serine specificity (Figure 3A). Based on their analysis of these five positions, they predicted that both glutamate and glycine receptors exist within each major lineage of metazoan iGluRs: AKDF, NMDA, epsilon, and lambda [11]. Their prediction certainly proved correct for the placozoan GluE1αA receptor characterized in this study (Figure 2A), as well as other non-vertebrate iGluRs that have been functionally characterized such as the glycine-sensitive AKDF1

receptor also from *T. adhaerens* [31], the glycine-activated GluE1 epsilon receptor from *Branchiostoma lanceolatum* [5], and the glutamate-activated kainate 1D receptor from *Drosophila melanogaster* [61]. To the best of our knowledge, ours is the first study to test this prediction model by seeking to switch ligand specificity of an iGluR”

Overall, the paper is clear, comprehensive, and ready for publication perhaps with the one minor adjustment above. I do not need to see the manuscript again before publishing.

Thank you